Letter

# Genome-wide association analyses identify distinct genetic architectures for early-onset and late-onset depression

John R. Shorter ®[1,25] ✉, Joëlle A. Pasman[2,3,25], Siim Kurvits[4,25],
Andreas Jangmo[5,25], Joonas Naamanka[6,7,25], Arvid Harder[2], Espen Hagen ®[8],
Kaarina Kowalec ®[2,9,10], Nelli Frilander[6], Richard Zetterberg[11],
Joeri J. Meijsen ®[11,12], Jesper R. Gådin[11], Jacob Bergstedt[13], Ying Xiong ®[2],
Sara Hägg ®[2], Mikael Landén ®[2,14], Christian Rück[15], John Wallert ®[15],
Alkistis Skalkidou ®[16], Elise Koch ®[8], Bayram C. Akdeniz[8], Oleksandr Frei ®[8],
FinnGen*, Iiris Hovatta ®[6], Ted Reichborn-Kjennerud ®[17,18],
Thomas M. Werge ®[11,19], Patrick F. Sullivan ®[2,20], Ole A. Andreassen ®[8,21],
Martin Tesli[18,22,23,24,26], Kelli Lehto[4,26], Alfonso Buil ®[11,26] & Yi Lu ®[2,26] ✉

Major depressive disorder (MDD) is a common and heterogeneous disorder of complex etiology. Studying more homogeneous groups stratified according to clinical characteristics, such as age of onset, can improve the identification of the underlying genetic causes and lead to more targeted treatment strategies. We leveraged Nordic biobanks with longitudinal health registries to investigate differences in the genetic architectures of early-onset (eoMDD; $n$ = 46,708 cases) and late-onset (loMDD; $n$ = 37,168 cases) MDD. We identified 12 genomic loci for eoMDD and two for loMDD. Overall, the two MDD subtypes correlated moderately (genetic correlation, $r_g$ = 0.58) and differed in their genetic correlations with related traits. These findings suggest that eoMDD and loMDD have partially distinct genetic signatures, with a specific developmental brain signature for eoMDD. Importantly, we demonstrate that polygenic risk scores (PRS) for eoMDD predict suicide attempts within the first 10 years after the initial diagnosis: the absolute risk for suicide attempt was 26% in the top PRS decile, compared to 12% and 20% in the bottom decile and the intermediate group, respectively. Taken together, our findings can inform precision psychiatry approaches for MDD.

Like other complex disorders such as type 2 diabetes[1] and epilepsy[2], the clinical heterogeneity observed in major depressive disorder (MDD) probably stems from the underlying etiological heterogeneity[3–5]. Recent advances in genome-wide association studies (GWAS) of MDD, facilitated by large samples from the Psychiatric Genomics Consortium[6,7], 23andMe[8], Million Veteran Program[9] and global biobanks[10–13], have yielded substantial progress in identifying MDD-associated genetic variants, and evidence of genetic differences between various clinical subtypes has emerged[3,7,13]. Despite these efforts, the quest for subtype-specific genetic loci in MDD has been slow, limiting our understanding of its complex etiology.

In this study, we investigate an important source of MDD heterogeneity, that is, age at onset (AAO), by stratifying individuals into early-onset MDD (eoMDD) and late-onset MDD (loMDD) and conducting a large GWAS meta-analysis of the two subtypes in the ongoing Nordic TRYGGVE collaboration[14,15] (Fig. 1). eoMDD is associated with

**Fig. 1 | Study design and analysis overview.** EMR, electronic medical record.

### Table 1 | Description of the cohorts used in this study

| Country | Study | Cohort type | n MDD cases | n MDD controls | n eoMDD cases | n eoMDD controls | n loMDD cases | n loMDD controls |
|---|---|---|---|---|---|---|---|---|
| Denmark | iPSYCH2012 | Case–cohort | 20,804 | 23,854 | 18,429 | 23,854 | NA | NA |
| | iPSYCH2015 | Case–cohort | 10,487 | 15,772 | 8,105 | 15,772 | NA | NA |
| Norway | MoBa | Population-based | 9,573 | 89,849 | 882 | 7,683 | NA | NA |
| Sweden | PREFECT, iCBT, STAGE, BASIC | Case–control | 4,107 | 19,416 | 470 | 3,711 | 483 | 3,711 |
| Estonia | EstBB | Population-based | 49,950 | 100,660 | 8,791 | 35,742 | 14,656 | 29,593 |
| Finland | FinnGen | Population-based | 56,661 | 113,322 | 10,031 | 20,062 | 22,029 | 88,116 |
| **Total** | | | **151,582** | **362,873** | **46,708** | **106,824** | **37,168** | **121,420** |
| UK (as comparison) | UKB | Population-based | 25,162 | 431,658 | 3,402 | 226,627 | 9,084 | 226,627 |

eoMDD (age at first MDD diagnosis ≤25 years). loMDD (age at first MDD diagnosis ≥50 years). The total sample sizes for the Nordic cohorts are shown in bold. EstBB, Estonian Biobank; iPSYCH, Integrative Psychiatric Research Consortium; MoBa, Norwegian Mother, Father and Child Cohort Study; UKB, UK Biobank.

severe outcomes, including psychotic symptoms, suicidal behavior and comorbidities with other mental disorders and somatic diseases[16,17], whereas loMDD tends to manifest with cognitive decline and increased cardiovascular disease risk[18]. Previous attempts to stratify MDD according to AAO have been hindered by methodological challenges, including large variations in AAO across samples, recall bias and relatively small sample sizes[17]. To address these challenges, we leveraged the Nordic biobanks and harmonized longitudinal health registries to stratify MDD cases based on age at first MDD diagnosis[15]. Previous research suggested that age at first diagnosis could be a useful proxy for AAO, given the high genetic correlation ($r_g = 0.95$) between the two phenotypes[16].

Following harmonization of phenotypic definitions of MDD and age at first diagnosis across nine cohorts from five Nordic countries (Denmark, Estonia, Finland, Norway and Sweden; Methods), we identified 151,582 MDD cases, including 46,708 eoMDD cases with age at first diagnosis of less than 25 years (approximating the 25th percentile of an AAO distribution ≤ 20–21 (ref. 19; Methods)) and 37,168 loMDD cases with age at first diagnosis of 50 years or older (approximating the 75th percentile of an AAO distribution ≥ 44–45 years; Methods) (Table 1)[19]. We conducted harmonized GWAS analyses on eoMDD and loMDD using singularity containers[20] in each cohort and then performed

meta-analyses. After phenotype harmonization, we observed high genetic correlations ($r_g = 0.7–0.9$) among the largest Nordic cohorts (the Integrative Psychiatric Research Consortium (iPSYCH), the Estonian Biobank (EstBB), FinnGen) (Supplementary Fig. 1). To assess generalizability outside the Nordic cohorts, we also analyzed UK Biobank (UKB) data, which relies on self-reported age at first diagnosis, and conducted GWAS of eoMDD and loMDD with the same age cutoffs (Supplementary Fig. 1). However, considering the major differences in samples and phenotypes, we conducted primary analyses based on Nordic cohorts of individuals with European ancestry, with the UKB as a comparison cohort for the identified loci.

We identified 12 genome-wide significant loci for eoMDD and two other loci for loMDD ($P < 5 \times 10^{-8}$) (Fig. 2). These loci were also captured in the GWAS of all cases with MDD, where we identified 80 significant loci (Supplementary Fig. 2 and Supplementary Table 1). Nearly all of these MDD loci have been reported in previous MDD GWAS[13] (Supplementary Table 3), demonstrating the validity of our harmonized phenotypes and GWAS. For the 17 significant genes identified in the eoMDD analysis, *BPTF*[21], *PAX5*[22], *SDK1*[23] and *SORCS3*[24] are involved in neurodevelopment or synaptic signaling (Supplementary Table 2). For loMDD, we identified four significant genes, with *BSN* implicated in synaptic neurotransmitter activity[25] (Supplementary Table 2).

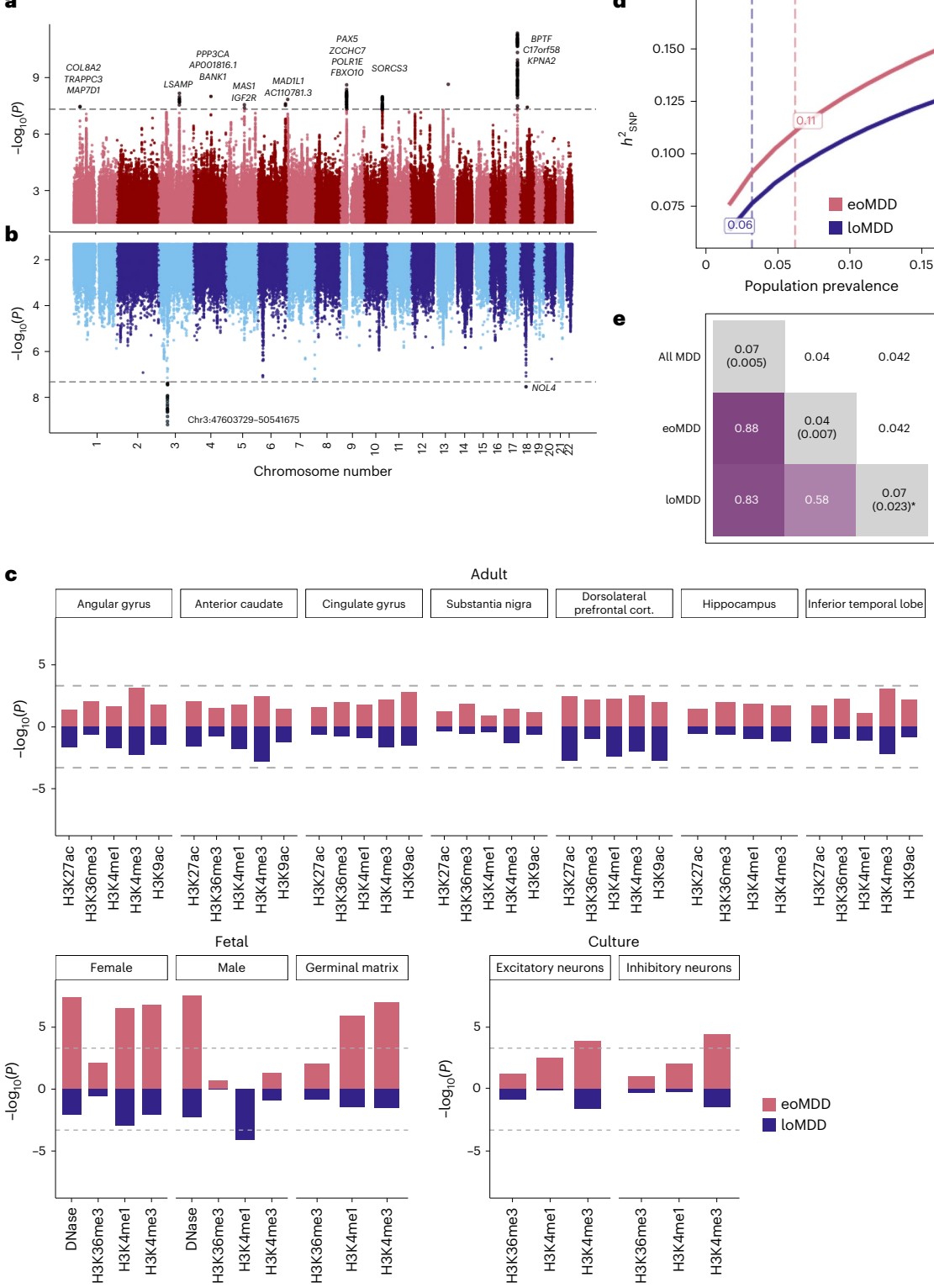

**Fig. 2 | MDD subtype GWAS meta-analysis in the Nordic cohorts and SNP heritability. a,b**, Mirrored Manhattan plots of the GWAS results from the combined Nordic countries, using inverse-variance-weighted meta-analysis with a genome-wide significance threshold of $P < 5 \times 10^{-8}$ (denoted by the dashed horizontal line), for eoMDD (**a**) and loMDD (**b**). **c**, Enrichment of open chromatin marks in the eoMDD and loMDD GWAS. The dashed line indicates the one-sided Bonferroni-corrected $P$ threshold, set at $P = 0.05/102 = 0.0005$. **d**, Single-nucleotide polymorphism (SNP)-based heritability $(h^2_{SNP})$ across a

range of population prevalence estimates, with the labeled dashed lines indicating the point estimate of the population prevalence. **e**, Genetic overlap between broad MDD, eoMDD and loMDD, with the upper triangle containing the standard errors and the diagonal containing estimates of polygenicity from SBayesS. * For loMDD, the algorithm did not converge, making the polygenicity estimate unreliable, although it was consistently higher than for eoMDD across successful runs.

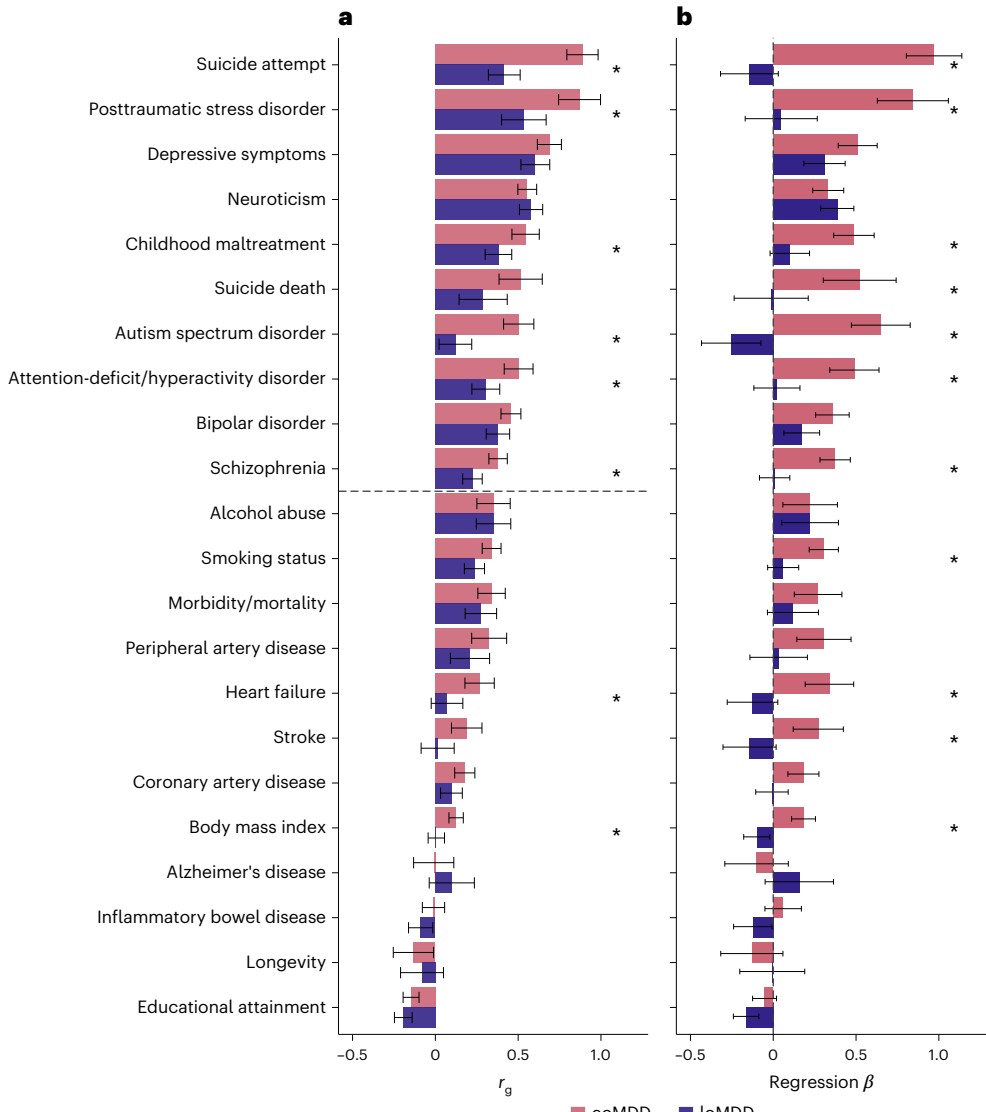

**Fig. 3 | Genetic correlations. a**, Genetic correlations from linkage disequilibrium (LD) score regression (LDSC) comparing eoMDD (based on GWAS $n$ = 153,532) and loMDD (GWAS $n$ = 158,588) with key health outcomes and related psychiatric disorders (GWAS $n$ in Supplementary Table 15). The error bars represent the 95% CIs. $P$ values for the associations are reported in Supplementary Table 8. **b**, Results from genomic SEM linear regression, where the outcomes are regressed on eoMDD (GWAS $n$ = 153,532) while controlling for overlap with loMDD (GWAS $n$ = 158,588), and vice versa. The single asterisks denotes a significant difference between eoMDD and loMDD at $P$ < 0.05. The error bars represent the 95% CIs.

Top loci were only partly replicated in the UKB, with only one locus on chromosome 9 showing nominal significance ($P$ < 0.05); however, for the genome-wide significant loci in either sample, the correlation in their effect sizes was substantial ($r$ = 0.84 for eoMDD; Supplementary Fig. 3 and Supplementary Tables 3–5).

By integrating GWAS findings on psychiatric disorders with tissue-specific open chromatin marks from the RoadMap Epigenomics Project[26], previous research implicated gene regulation during fetal neurodevelopment[27]. We tested the specific hypothesis that the GWAS of eoMDD would be enriched at regulatory chromatin marks active in fetal brains. Indeed, we found that eoMDD genetic signals were significantly enriched in fetal brain tissues, whereas no enrichment was detected in adult brains (Fig. 2c and Supplementary Table 6a), suggesting a role of early brain development in the risk of eoMDD. For loMDD, we only detected enrichment of one epigenetic marker in male fetal tissues. We did not identify significant enrichment in Genotype-Tissue Expression (GTEx) brain tissues[28] or human adult brain cell types[29], potentially because of the relatively low statistical

power in these subtype-specific GWAS (Supplementary Fig. 4 and Supplementary Table 6b,c).

The single-nucleotide polymorphism (SNP)-based heritability $\left(h^2_{SNP}\right)$ estimates for eoMDD were higher than for loMDD at a range of population prevalences (Fig. 2d and Supplementary Table 7). Assuming a population prevalence of 6.2% (Methods), the $h^2_{SNP}$ for eoMDD was estimated at 11.2% (95% confidence interval (CI) = 9.9–12.5%; liability scale), which was almost twice as high as that of loMDD at 6% (95% CI = 4.3–6.3%) at a prevalence of 3.2% (Supplementary Table 7). We also conducted a case–case GWAS directly comparing eoMDD with loMDD; however, this comparison showed a small $h^2_{SNP}$ (2%, s.e. = 0.74%). Interestingly, analyses of the genetic architecture using SBayesS[30] estimated that the polygenicity (that is, the proportion of SNPs with nonzero effects) in eoMDD (4%, posterior s.e. = 0.7%) was much lower than our GWAS of all MDD cases (7%, posterior s.e. = 0.5%) (Fig. 2e) and the estimate reported in the latest Psychiatric Genomics Consortium MDD GWAS (6%)[6], suggesting that fewer causal variants underpin eoMDD.

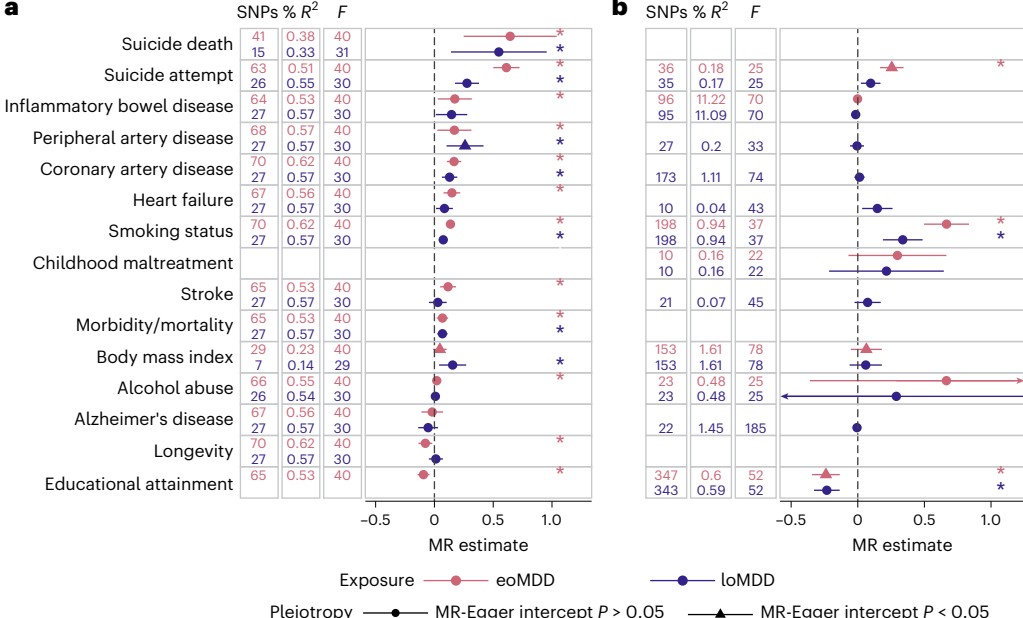

**Fig. 4 | Causal relationships between eoMDD, loMDD and health outcomes. a**, Results of the MR analyses with MDD as the exposure, with the IVW statistic as the MR effect size estimate (ordered according to the eoMDD effect size). The error bars represent the 95% CIs. **b**, Effects with MDD as the outcome (ordered according to the effect size in **a**). Included in both **a** and **b** are the number of instrument SNPs used in the analysis ('SNPs'), the percentage variance explained by the instrument SNPs in the exposure ('%$R^2$') and the instrument strength ('$F$'). We only tested plausible relationships (in line with time ordering), with improbable relationships, for example, suicide death as a risk factor for MDD, excluded from the figure. If the MR-Egger sensitivity analysis had a significant intercept (triangles), this indicates that the MR estimate was pleiotropic and should not be interpreted. Estimates with an asterisk were significant after correction for multiple testing (exact $P$ values are reported in Supplementary Table 10). The error bars represent the 95% CIs.

The two MDD subtypes correlated moderately ($r_g = 0.58$, s.e. = 0.04) (Fig. 2e and Supplementary Table 8) and showed differences in their genetic correlations with other traits (Fig. 3a, Supplementary Fig. 5 and Supplementary Table 8). eoMDD had the highest $r_g$ with suicide attempt ($r_g = 0.89$, s.e. = 0.05), which was more than twice the $r_g$ between loMDD and suicide attempt ($r_g = 0.42$, s.e. = 0.05) (comparison with $r_g$ in all MDD in Supplementary Fig. 6 and Supplementary Table 8). Similarly, substantial differences were found in their $r_g$ with posttraumatic stress disorder, childhood maltreatment, attention-deficit/hyperactivity disorder, autism spectrum disorder and schizophrenia. For somatic and lifestyle phenotypes, the overlap with eoMDD was significantly stronger for heart failure and body mass index (Supplementary Table 8). Given the $r_g$ between the two MDD subtypes, we further investigated the independent genetic effects of each subtype while conditioning on the other using genomic structural equation modeling (SEM) (Fig. 3b). In this way, the unique overlap between each subtype and the other traits could be evaluated, giving insight into how much of the overlap with the other trait was driven by overlap with the other subtype. After controlling for loMDD, the genetic associations of eoMDD with other traits remained similar, except that the negative $r_g$ between eoMDD and educational attainment was attenuated (Supplementary Table 9). On the other hand, after accounting for the genetic overlap with eoMDD, the genetic associations of loMDD were substantially reduced for many traits (for example, suicide attempt and suicide death), suggesting that the observed $r_g$ between loMDD and these traits were driven by the shared genetics with eoMDD. Overall, eoMDD had stronger genetic overlaps with psychiatric and general health traits than loMDD, with the most notable difference in their genetic correlations with suicide attempt.

We further investigated these relationships using two-sample Mendelian randomization (MR) (Fig. 4). Considering the clear timing difference in the MDD subtypes, we were primarily interested in the role of eoMDD as a risk factor for general health outcomes, while providing the estimates of loMDD as a comparison, and the role of general health traits with loMDD as an outcome. Of note, eoMDD showed a significantly larger effect on suicide attempt than loMDD (eoMDD $\beta = 0.61$, s.e. = 0.057; loMDD $\beta = 0.28$, s.e. = 0.052), while the magnitude of effects on suicide death was more comparable between the two subtypes (eoMDD $\beta = 0.64$, s.e. = 0.20; loMDD $\beta = 0.55$, s.e. = 0.21). For educational attainment, our results were consistent with a previous report that suggested lower educational attainment as a putatively causal risk factor for MDD[7], in particular for loMDD ($\beta = -0.23$, s.e. = 0.049; the effect on eoMDD was pleiotropic); furthermore, we showed that eoMDD had a putatively causal effect on lower educational attainment ($\beta = -0.09$, s.e. = 0.026). Similarly, our results that both subtypes had a small putatively causal effect on cardiovascular disease (CVD) were in line with previous findings considering MDD as a whole[31]; there was evidence suggesting that heart failure was putatively causal for loMDD. Results from sensitivity analyses were consistent with these main findings (Supplementary Table 10).

To investigate whether polygenic risk scores (PRS) for eoMDD and loMDD were associated with clinical indicators or severe outcomes, we generated PRS using leave-one-out (LOO) GWAS summary statistics and tested PRS associations with these outcomes extracted from longitudinal health registry data in each cohort. In the meta-analysis and across individual cohorts, the eoMDD PRS explained a greater proportion of the phenotypic variance compared to the loMDD PRS for all outcomes of interest (Fig. 5a, Supplementary Fig. 7 and Supplementary Table 11). In particular, a 1 s.d. increase in eoMDD PRS was associated with an increased risk of early-onset (odds ratio (OR) = 1.26, 95% CI = 1.13–1.4, $P = 1.8 \times 10^{-5}$) and lifetime risk of MDD (OR = 1.24, 95% CI = 1.17–1.31, $P = 1.16 \times 10^{-13}$) compared to the loMDD PRS for early (OR = 1.13, 95% CI = 1.08–1.17, $P = 3.9 \times 10^{-9}$) and lifetime risk of MDD (OR = 1.13, 95% CI = 1.07–1.19, $P = 1.4 \times 10^{-5}$). Similarly, compared to the loMDD PRS, the eoMDD PRS was more strongly associated with other outcomes of interest; differences in estimates for eoMDD and loMDD PRS were

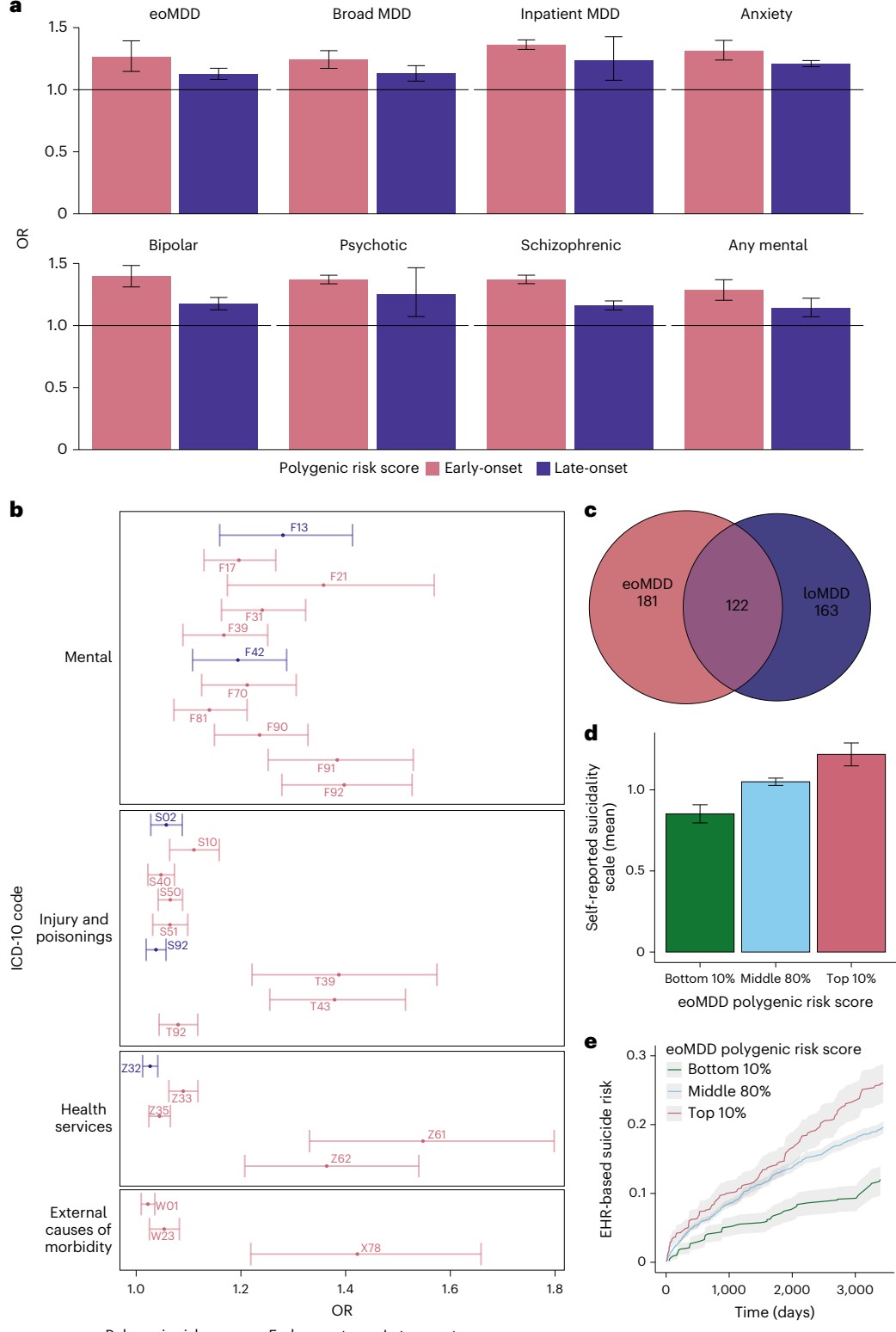

**Fig. 5 | Associations of PRS for eoMDD and loMDD with clinical outcomes and suicide. a**, Meta-analyzed associations between PRS and MDD outcomes in the Nordic cohorts, using LOO GWAS summary statistics of eoMDD and loMDD. The error bars represent the 95% CIs. **b**, Selected mental-health-related and suicide-related ICD-10 group associated with eoMDD and loMDD from the PRS PheWAS. The error bars represent the 95% CIs. **c**, Number of unique and shared associations with eoMDD and loMDD from the PheWAS. **d**, Mean symptom scores of self-reported suicidality (Paykel Suicide Scale) stratified according to the top 10%, middle 80% and bottom 10% eoMDD PRS. The error bars represent the 95% CIs. **e**, Cumulative incidence of suicide attempt over a 10-year period since the first eoMDD diagnosis, stratified according to the top, middle and bottom eoMDD PRS. The gray shading indicates the 95% CIs.

particularly pronounced for hospitalization and diagnostic conversion to bipolar disorder and schizophrenia.

To understand how genetic predispositions underlying the two subtypes are associated with comorbidities, we conducted a PRS-based phenome-wide association study (PheWAS) using the EstBB medical records from both primary and specialist care (1,428 International Classification of Diseases, Tenth Revision (ICD-10) code-based diagnosis). After correction for multiple testing (Bonferroni correction, P threshold of 0.05/1,428), we identified both shared and unique associations with either subtype (Fig. 5c and Supplementary Fig. 8a,b). Notably, the eoMDD PRS showed unique associations with psychiatric conditions such as conduct disorder (ICD-10 code: F91) and schizotypal disorder (F21), suicide attempt/intentional self-harm (X78) and problems related to negative life events in childhood (Z61), or other problems related to upbringing (Z62). In contrast, the loMDD PRS was more notably associated with mental and behavioral disorders because of the use of sedatives or hypnotics (F13) and obsessive–compulsive disorder (F42) (Fig. 5b and Supplementary Table 12).

Prompted by our findings of the strong genetic link between eoMDD and suicide attempt, we investigated whether the eoMDD PRS predicts the risk of suicide attempt using self-report surveys and medical records from EstBB. We stratified individuals with eoMDD into three subgroups based on the eoMDD PRS (top and bottom deciles, and middle 80%). First, we observed a dose–response relationship between eoMDD PRS strata and the mean scores of the self-reported Paykel Suicide Scale[32], that is, the higher eoMDD PRS, and the higher symptoms of suicide thoughts and attempts (mean symptoms scores = 0.85, 95% CI = 0.79–0.91; 1.05 (1.03–1.07), 1.22 (1.15–1.29) for the bottom, middle and top PRS strata, respectively) (Fig. 5d and Supplementary Table 13a). Next, we estimated the hazard risk ratio (HRR) and the absolute risk for suicide attempt treated in primary or specialist care within a 10-year period since the first recorded MDD diagnosis. Compared with the middle PRS group, individuals in the lowest PRS decile had a significantly lower HRR for suicide attempt of 0.57 (95% CI = 0.49–0.68, $P = 1.34 \times 10^{-10}$), while those in the highest PRS decile had an increased risk (HRR = 1.13, 95% CI = 1.00–1.28, P = 0.058). Furthermore, individuals with the lowest PRS showed a consistently low absolute risk of suicide attempt over time after the first MDD diagnosis (12% cumulative incidence over a 10-year period), while there were no major differences in absolute risk between the middle and highest PRS group until 5.5 years, with the 10-year cumulative incidence at 20% and 26%, respectively (Fig. 5e).

Previous research showed that the incidence of suicide attempts is particularly pronounced during adolescence and young adulthood[33]. Early identification of individuals at heightened risk for suicide is of major clinical importance. This period of high incidence of suicide attempts coincides with the risk for eoMDD, making it particularly relevant to perform risk prediction of suicide attempts in this subgroup. Together with our findings suggesting that eoMDD is putatively causal for suicide attempts, these results underscore the potential use of eoMDD PRS in stratifying the risk of suicide attempt among individuals with eoMDD (Supplementary Table 13b) and warrant further investigation of its relevance for suicide prevention[33].

MDD genomics has achieved tremendous progress, from zero significant associations in 2013 to 697 associations in 2025 for the MDD case–control phenotype[6,7,34]. In this study, we explored the strategy of going beyond case–control to target specific phenotypic subgroups, aiming to reduce genetic heterogeneity in MDD. Our harmonized healthcare data and analyses across countries improved analytical power and identified differential genetic signals in AAO-based subtypes. A similar approach can be extended to other clinical characteristics, such as vegetative symptoms, psychotic features and disability, to further target clinically relevant subtypes of MDD[3,5,35,36]. These findings may have important implications for guiding targeted treatment and prevention in psychiatry.

## Online content

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

[1]Department of Science and Environment, Roskilde University, Roskilde, Denmark. [2]Department of Medical Epidemiology and Biostatistics, Karolinska Institutet, Stockholm, Sweden. [3]Department of Psychiatry, Amsterdam UMC location University of Amsterdam, Amsterdam, the Netherlands. [4]Estonian Genome Centre, Institute of Genomics, University of Tartu, Tartu, Estonia. [5]Department of Chronic Diseases, Norwegian Institute of Public Health, Oslo, Norway. [6]SleepWell Research Program and Department of Psychology, Faculty of Medicine, University of Helsinki, Helsinki, Finland. [7]Central Institute of Mental Health, Hector Institute for Artificial Intelligence in Psychiatry, Mannheim, Germany. [8]Centre for Precision Psychiatry, Institute of Clinical Medicine, University of Oslo and Oslo University Hospital, Oslo, Norway. [9]College of Pharmacy, University of Manitoba, Winnipeg, Manitoba, Canada. [10]Department of Biochemistry and Medical Genetics, University of Manitoba, Winnipeg, Manitoba, Canada. [11]Institute of Biological Psychiatry, Mental Health Center Sct. Hans, Mental Health Services Copenhagen, Copenhagen University Hospital, Roskilde, Denmark. [12]Center for Eating and feeding Disorders Research, Mental Health Center Ballerup, Copenhagen University Hospital, Mental Health Services Copenhagen, Copenhagen, Denmark. [13]Unit of Integrative Epidemiology, Institute of Environmental Medicine, Karolinska Institutet, Stockholm, Sweden. [14]Institute of Neuroscience and Physiology, University of Gothenburg, Gothenburg, Sweden. [15]Centre for Psychiatry Research, Department of Clinical Neuroscience, Karolinska Institutet & Stockholm Health Care Services, Region Stockholm, Karolinska Institutet, Stockholm, Sweden. [16]Department of Women's and Children's Health, Uppsala University, Uppsala, Sweden. [17]Department of Mental Health and Suicide, Norwegian Institute of Public Health, Oslo, Norway. [18]Institute of Clinical Medicine, University of Oslo, Oslo, Norway. [19]Department of Clinical Medicine, Faculty of Health Sciences, University of Copenhagen, Copenhagen, Denmark. [20]Departments of Genetics and Psychiatry, University of North Carolina at Chapel Hill, Chapel Hill, NC, USA. [21]KG Jebsen Centre for Neurodevelopmental Disorders, University of Oslo, Oslo, Norway. [22]Division of Mental Health and Substance Abuse, Diakonhjemmet Hospital, Oslo, Norway. [23]Department of Mental Health, Norwegian Institute of Public Health, Oslo, Norway. [24]Department of Psychiatry, Østfold Hospital, Grålum, Norway. [25]These authors contributed equally: John R. Shorter, Joëlle A. Pasman, Siim Kurvits, Andreas Jangmo, Joonas Naamanka. [26]These authors jointly supervised this work: Martin Tesli, Kelli Lehto, Alfonso Buil, Yi Lu. *A list of authors and their affiliations appears at the end of the paper. ✉e-mail: johnsh@ruc.dk; lu.yi@ki.se

## FinnGen

### Iiris Hovatta[6]

A full list of members and their affiliations appears in the Supplementary Information.

## Methods

### Study population

To conduct a large GWAS of AAO-based MDD subtypes with consistent phenotypes, we identified nine cohorts across five Nordic countries (Denmark, Sweden, Norway, Finland and Estonia), and a comparison cohort from a non-Nordic country (the UK: UKB). Many of these cohorts are large biobanks linked with lifetime medical records in national patient registers, including iPSYCH, FinnGen, the Norwegian Mother, Father and Child Cohort Study (MoBa), the UKB and the EstBB (cohort details in Supplementary Methods). This study was approved by the Swedish ethical review authority (case no. 2023-03073), and each cohort was approved by the relevant institutional review boards.

### Phenotypes

We expanded our previous effort of harmonizing the register-based phenotypes of MDD, age at first diagnosis and its outcomes in the three Scandinavian countries to other Nordic countries with similar registers[15]. Using national patient registers, we first extracted information on patient diagnoses of MDD using ICD-10 code F32 (depressive episode) or F33 (recurrent depressive episodes), with the exclusion criteria of a lifetime diagnosis of bipolar disorder or schizophrenia (ICD-10 codes: F30/F31/F32/F33, F20/F23.1/F23.2/F25) (Supplementary Table 14), resulting in a total combined number of MDD cases in the Nordic countries of $n = 151,582$. Our previous research revealed a high genetic correlation ($r_g \approx 0.95$) between AAO and age at first diagnosis for MDD[16], and that onset on average predates first diagnosis by 5 years[16]. Therefore, we derived the age at first diagnosis for the MDD patient population and used that as a proxy for AAO for the definition of eoMDD and loMDD. The cutoffs were chosen based on careful review of the literature and the empirical registry data from Sweden. Previous meta-analysis reported that the median AAO for depressive disorder is around age 30, with the 25th and 75th percentile at age 21 and 44, respectively[15,19]; our Swedish registry data showed that for age at first MDD specialist diagnosis, the 25th and 75th percentile were at age 25 and 45, respectively[15]. Thus, we considered individuals with their first psychiatric specialist treatment contact for MDD at or before age 25 (approximate to the 25th percentile of the AAO distribution at age 20–21) as cases with eoMDD ($n = 46,708$); cases with loMDD were those with their first specialist treatment contact for MDD at or after age 50 (approximate to the 25th percentile of the AAO distribution at age 44–45; $n = 37,168$). Controls were individuals without a registered diagnosis of MDD, bipolar disorder or schizophrenia. Because of the differences in cohort design, controls were matched to cases for the population-based cohorts of EstBB, FinnGen, MoBa and UKB. The Danish and Swedish cohorts, which used case–cohort and case–control designs, did not require control matching.

### Software containers for federated analyses

To ensure transparent and reproducible analyses of data across sites, we developed software containers and accompanying codes for data processing and analyses (for example, GWAS, PRS, LDSC) and distributed the containers across the study sites. Briefly, the primary software tools and dependencies were installed in virtual machines based on the Ubuntu 20.04 LTS Linux operating system via Docker (https://docker.com) and containerized using the Singularity Image File format (https://sylabs.io) for distribution[20]. All source codes and files are publicly available via GitHub and released under the GNU General Public License (GPL v.3.0). Tools released under other licenses retain their original licenses.

### GWAS and meta-analysis

GWAS of individuals with European ancestries was conducted in each cohort using the REGENIE software package[37] available in the software containers[20]. Analyses were adjusted for the first ten principal components (PCs), age and sex. Summary statistics from each cohort were included in a fixed-effects meta-analysis in METAL[38] with the 'SCHEME STDERR' command using the inverse variance weight of the corresponding standard errors for all phenotypes studied, resulting in 151,582 cases and 362,873 controls for all MDD, 46,708 cases and 106,824 controls for eoMDD, and 37,168 cases and 121,420 controls for loMDD. Additional variant filtering was applied: INFO score > 0.8, minor allele frequency ≥ 1% and $n > 10,000$. After this filtering, the numbers of markers for the final METAL analyses were 8,910,578 for broad MDD, 8,848,589 for eoMDD and 8,820,060 for loMDD. Genome-wide significance was set at $P < 5 \times 10^{-8}$. Genome-wide loci were identified based on genome-wide significant SNPs using PLINK (--clump command). A gene list was then created based on which genes physically overlapped the loci from the previous step (Supplementary Table 2). An additional case–case GWAS was performed comparing eoMDD cases to loMDD cases using the EstBB and FinnGen cohorts, as they include both eoMDD and loMDD cases in their biobanks. A total of 32,060 samples from FinnGen and 23,447 samples from EstBB were meta-analyzed using METAL; heritability ($h^2_{SNP}$) was estimated using LDSC[39,40].

### Gene-based tests

We used MAGMA[41] v.1.08 to test the aggregate association of variants by assigning SNPs to genes. Gene boundaries were expanded by 35 kb upstream and 10 kb downstream to allow for the inclusions of promoters and cis-enhancers. We used the updated deep whole-genome sequencing of the 1000 Genomes Project (European subset) as reference data[42], downloaded from http://ftp.1000genomes.ebi.ac.uk/vol1/ftp/data_collections/1000G_2504_high_coverage/working/20220422_3202_phased_SNV_INDEL_SV/.

### Tissue-type and cell-type enrichment of SNP heritability with stratified LDSC

We used stratified LDSC[39] to perform tissue-type and cell-type analysis, estimating the enrichment of SNP heritability in a set of genome annotations. We downloaded precomputed LDSC scores from https://console.cloud.google.com/storage/browser/_details/broad-alkesgroup-public-requester-pays/LDSCORE/LDSC_SEG_ldscores/biorxiv/Roadmap_1000Gv3.tgz, containing data on 396 annotations of epigenetic marks from the ROADMAP project, restricting to the 102 annotations relating to the brain. For the gene expression data, we followed our previous protocol[43], first aggregating gene expression across genes to get the average expression per gene in each cell type. We then defined cell-type specificity by dividing the expression of a gene in a cell type by its total expression across all cell types. For each cell type, we considered the top decile of specifically expressed genes as typifying that cell type, and computed LDSC scores for that annotation. In addition to the ROADMAP annotations, we also tested the enrichment of SNP-based heritability (one-sided) in human tissues (37 human tissues from GTEx[28]) and brain cell types using the latest human single-nucleus transcriptomic data (31 major human brain cell types from ref. 29). We applied a Bonferroni correction for multiple testing in each annotation and trait (for example, in the analysis of ROADMAP data, significance was set at $P < 0.05/102$ for each trait).

### SNP-based heritability and assessing genetic overlap

We used LDSC to estimate SNP-based heritability and genetic correlation[40]. To convert the SNP-based heritability to the liability scale, we calculated the sample prevalence of cases in the meta-analysis; for population prevalence, we used a lifetime estimate of diagnosed MDD of 16% (ref. 44). To obtain corresponding estimates for eoMDD and loMDD, we estimated the proportion of cases with MDD who had an early-onset or a late-onset diagnosis in the largest population samples (EstBB and FinnGen) and took that as a percentage of the whole MDD prevalence estimate, resulting in a population estimate of 6.2% for eoMDD and 3.2% for loMDD. Acknowledging that these estimates are

approximations, we provided SNP heritability estimates for a range of prevalence estimates (±50% around the point estimates).

To estimate genetic correlations with other traits, we used phenotypes from publicly available summary statistics, selecting important psychiatric disorders and key health determinants and outcomes, including suicide, body mass index, educational attainment, substance use, mortality and cardiovascular disease (GWAS sources in Supplementary Table 15).

## SBayesS
Using SBayesS[30], we confirmed the LDSC estimates of SNP-based heritability and further estimated the parameter of polygenicity, that is, the proportion of SNPs with nonzero effects, and negative selection. We used the precomputed LD Matrix from UKB participants ('ukbEURu_imp_v3_HM3_n50k.chisq10'), which is available from the software authors. We ran four chains in parallel to estimate the convergence across starting parameters (--num-chains 4), with each run consisting of 25,000 iterations (--chain-length 25,000), with 5,000 iterations considered as burn-in iterations (--burn-in 5,000).

## Genomic SEM
We used genomic SEM to compare the extent to which genetic liability to eoMDD versus loMDD overlaps with genetic liability to other phenotypes[45]. Genomic SEM is an extension of SEM whereby GWAS summary statistics are used to capture the 'observed' traits in the model so that the genetic architecture underlying the multivariate relationships between traits can be modeled. Genomic SEM relies on LDSC to estimate genetic correlations and is robust to any amount of sample overlap. We modeled the association of the MDD subtypes with other related phenotypes that have been well studied in the literature, including a range of psychiatric disorders, important health determinants, CVD and mortality (Supplementary Table 15). First, we compared the size and significance of the genetic correlations of the two subtypes (eoMDD and loMDD) with the other phenotypes. Next, we assessed the subtype-specific contribution to the other phenotypes by controlling for the other subtype (for example, we investigated the relationship between eoMDD and suicide attempt after controlling for loMDD). Thus, the unique contribution of each can be gauged while controlling for their shared variance. The genetic overlap between eoMDD and loMDD was moderate; therefore, the concern for strong bias through multicollinearity was limited.

## Mendelian randomization
Following the analyses of genetic overlap, we used MR to assess whether the MDD subtypes were causally associated with selected health outcomes (risk factors, mortality and CVD; Supplementary Table 15). We did not include psychiatric outcomes because of the high likelihood of pleiotropy. Assuming that alleles are randomly distributed in the population (in a manner akin to experimental randomization), MR uses genetic variants as instrumental variables to capture a trait ('exposure'). If the genetic variants are robust, non-pleiotropic instruments for the exposure and there is no direct association with the outcome or an unmeasured confounder, this provides support for a potential causal effect of the exposure. We conducted main and sensitivity analyses as implemented in the TwoSampleMR R package[46] and computed the IVW statistic to estimate these associations. For interpretation, we focused on effect consistency across tests, rather than $P$ values. Still, to give an indication of the extent of the multiple testing burden, we report significance according to false discovery rate-corrected $P$ values (Benjamini–Hochberg method). We present several sensitivity analyses that correct for potential violations of the MR assumptions. Weighted median and mode analyses are more robust to invalid instruments and correct for outliers[47]. The MR-Egger analysis was used to assess any remaining pleiotropy (intercept) and adjust for such effects. The MR-Egger estimate is often underpowered, but it is still informative to

check whether the effect is in the same direction as the other sensitivity analyses[48]. From the same TwoSampleMR package, it is possible to estimate the amount of heterogeneity ($Q$) in the SNP effects, and estimate the instrument strength $F$. We added a sensitivity analysis from the MR-PRESSO R package[49], which has a different and more powerful way to adjust for the effects of outliers and is aimed at correcting for pleiotropy. Finally, we assessed the 'no measurement error' assumption by deriving the $I^2$ statistic and applied a SIMEX correction on the MR-Egger estimate if $I^2$ was between 0.6 and 0.8 (we did not present the MR-Egger estimate when the $I^2$ fell below the 0.6 cutoff[50]). Although we were not able to remove potential sample overlap from the analyses (LOO analyses would leave the exposure GWAS underpowered), this should not be a large threat to the validity of our results. The extent of bias stemming from sample overlap is limited if the sample size of the source GWAS is large, the overlap is small and the heritability of the investigated traits is high[51]. As shown in Supplementary Table 15, the amount of sample overlap was limited in most cases and present only for a subset of traits.

The aim of the MR analyses was to contrast causal associations between eoMDD and loMDD. However, there was overlap in the instrument sets for both traits. Although only one SNP is present in both sets, many had LD partners in common. Therefore, we performed another sensitivity analysis using a single instrument SNP for each trait. We selected the genome-wide significant SNP that had the largest $P$ value for the other trait (filtering on a minor allele frequency > 0.10). For eoMDD, we used rs7622302 residing in the *DAG1* gene on chromosome 3, with $P = 0.98$ for loMDD. This instrument explained 2.8% of the variance ($R^2$) in eoMDD. For loMDD, we used rs3777421 in the *IGF2R* gene on chromosome 6, with $P = 0.87$ for eoMDD. This SNP explained 2.7% variance in loMDD. Although power was expected to be lower in these single-SNP analyses, pleiotropy was also reduced; instrument strength was still sufficient ($F = 30.7$ for eoMDD and $F = 30.1$ for loMDD) and the contrast between eoMDD and loMDD was maximized.

## Polygenic risk scores
PRS were calculated using LDpred2[52], which addresses some limitations of LDpred[53]. Briefly, LDpred2 is a Bayesian method that uses the LD structure in the genome combined with priors regarding the genetic architecture of a trait (SNP heritability and fraction of causal SNPs (hyperparameter p)) to compute PRS. We used the 'auto' method of LDpred2 that maximizes the predictive performance of the PRS by testing a range of the fraction of causal SNPs. For our analyses, we used LD matrices provided by the authors of LDpred2[52], which were generated from the UKB samples (European ancestry) with 1.1 million HapMap3 SNPs. The LDpred2 calculations relied on the bigsnpr R library[54] and were implemented in a set of custom R scripts[55].

Individual-level PRS in each cohort were computed using the LOO GWAS summary statistics on MDD subtypes and tested for associations with clinical characteristics or MDD-related outcomes (Supplementary Table 16) in logistic regression models adjusted for at least ten PCs (cohort-dependent), sex and birth year. For the outcomes that were available in all target samples, we performed fixed-effects meta-analysis using the R package metafor[56].

## PRS PheWAS
To investigate the genetic comorbidity of eoMDD and loMDD, we conducted a PRS PheWAS using logistic regression models, using the EstBB electronic health record (EHR) data. Analyses were performed in R (v.4.3). For each ICD-10 code with at least 50 cases ($n = 1,151$) present in the EstBB EHR, two models were constructed—eoMDD PRS and loMDD PRS as the predictor. These models used the normalized PRS (either eoMDD or loMDD) and a binary outcome variable indicating the presence or absence of the ICD-10 code. The EHR data encompassed both primary and secondary/inpatient care data through linkage to the Estonian central e-health database, covering the time from 2004 to 2022.

The models included sex, year of birth and the first ten genetic PCs as covariates. To account for multiple testing, we applied Bonferroni correction, resulting in a conservative *P* threshold of 0.05/1,151. ORs (with 95% CIs), the number of cases per ICD-10 code and the *P* values for both eoMDD and loMDD models are provided in Supplementary Table 12 and Supplementary Figure 8a,b.

### Self-reported suicidality stratified according to eoMDD PRS

To evaluate the effect of the eoMDD PRS on self-reported suicidality, we used data from the EstBB Mental Health Online Survey[57]. Self-reported suicidality was assessed using the Paykel Suicide Scale, which includes five items assessing self-reported suicidal thoughts and attempts[32]. A total of 58,732 individuals who had an MDD diagnosis were present in the EHRs (F32 or F33 codes) and responded to the Paykel Suicide Scale. For this analysis, the first four questions were scored on a scale from 0 to 4 (0 indicating the participant had never felt or experienced this, and 4 indicating the participant felt or experienced it often). The fifth question had binary responses (no/yes) corresponding to values of 0 or 2. The total score of the five items was used for the analysis; the distribution according to eoMDD PRS group is provided in Supplementary Table 13a. The mean values and standard deviations of these sums were then computed for three groups based on eoMDD PRS deciles (1st, 2nd to 9th, 10th). Analyses were conducted using R (v.4.3).

### Cox proportional-hazards model of suicide attempt stratified according to eoMDD PRS

To estimate the absolute risk of EHR-based suicide attempt over 10 years after the first diagnosis of MDD, stratified according to eoMDD PRS, we used a Cox proportional-hazards model[58], using data from the EstBB (results in Supplementary Table 13). Sex and year of birth were included as covariates. The analysis included a cohort of 10,539 individuals diagnosed with MDD (first diagnosis date in 2000 or later, identified by the ICD-10 codes F32 or F33, before or at the age of 25). Participants' birth years ranged from 1974 to 2004, with a mean birth year of 1990.

We calculated the absolute risk (with 95% CIs) of suicide attempt after the first episode of depression for three groups based on eoMDD polygenic risk score deciles: the 1st decile, the 2nd to 9th deciles and the 10th decile. The risk of suicide attempt was assessed over a 10-year period (3,650 days) after the initial MDD diagnosis. ICD-10 codes X60-X84 and Y87 from the Estonian EHRs were used to identify suicide attempts. The absolute risk was presented as 1 minus the survival probability. All analyses were conducted using R v.4.3, with the ggsurvfit (v.1.0.0)[58] and survival (v.3.5.7) packages[59].

### Reporting summary

Further information on research design is available in the Nature Portfolio Reporting Summary linked to this article.

## Data availability

The GWAS summary statistics reported in this article are available via figshare at https://doi.org/10.6084/m9.figshare.27830340 (ref. 60).

## Code availability

The central code repository for the software containers for the GWAS and post-GWAS analyses is available via GitHub at https://github.com/comorment/containers and Zenodo at https://doi.org/10.5281/zenodo.15096746 (ref. 55); the other tools and public reference data are available via GitHub at https://github.com/comorment/containers.

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

## Acknowledgements

This work was supported by the US National Institutes of Mental Health (R01s MH124871, MH121545 and MH123724 to P.F.S.); the European Union's Horizon 2020 Research and Innovation Programme (grant nos. 847776 (CoMorMent) and 964874 (RealMent) to O.A.A.); the European Research Council (grant no. 101042183 to Y.L.); the Research Council of Norway (grant nos. 324499 and 296030 to O.A.A.); NordForsk (no. 164218 to O.A.A.); the Swedish Research Council (award no. D0886501 to P.F.S., award no. 2021-02615 to Y.L., award no. 2021-06377 to J.W. and award no. 2022-01643 to M.L.); the Swedish Brain foundation (no. FO2022-0217 to M.L.); the Swedish Research Council for Health, Working Life and Welfare (agreement no. 2022-00814 to J.A.P.); the Estonian Research Council (no. PSG615 to K.L.); and the Estonian Centre of Excellence for Well-Being Sciences, funded by grant no. TK218 from the Estonian Ministry of Education and Research (to K.L.). We acknowledge the Estonian Biobank (EstBB) research team: A. Metspalu, L. Milani, T. Esko, R. Mägi, M. Nelis and G. Hudjashov. Research in the EstBB was supported by the European Union through the European Regional Development Fund project no. 2014-2020.4.01.15-0012 GENTRANSMED. Data analysis in the EstBB was carried out in part at the High-Performance Computing Center of the University of Tartu. The computations for the STAGE cohort and the UKB were enabled by resources provided by the National Academic Infrastructure for Supercomputing in Sweden and the Swedish National Infrastructure for Computing at the Uppsala Multidisciplinary Center for Advanced Computational Science, which is partially funded by the Swedish Research Council through grant agreement nos. 2022-06725 and 2018-05973. This study used computing resources provided by UNINETT Sigma2 at the University of Oslo (UIO/USIT/TSD p697, grant no. NS9703S). The PREFECT study was funded by the Swedish foundation for Strategic Research (no. KF10-0039). The BASIC study was supported by the Swedish Research Council (case no. 523-2014-2342). Analyses in the UKB have been conducted under application no. 22224. The MoBa Study is supported by the Norwegian Ministry of Health and Care Services and the Ministry of Education and Research. We thank all the participating families in Norway who take part in this ongoing cohort study. The FinnGen project is funded by two grants from Business Finland (HUS 4685/31/2016 and UH 4386/31/2016) and the following industry partners: AbbVie, AstraZeneca UK, Biogen MA, Bristol Myers Squibb (and the Celgene Corporation & Celgene International II Sàrl), Genentech, Merck Sharp & Dohme, Pfizer, GlaxoSmithKline Intellectual Property Development Ltd., Sanofi US Services, Maze Therapeutics, Johnson & Johnson Innovative Medicine, Novartis, Boehringer Ingelheim and Bayer. The following biobanks are acknowledged for delivering biobank samples to FinnGen: Auria Biobank (www.auria.fi/biopankki), THL Biobank (www.thl.fi/biobank), Helsinki Biobank (www.helsinginbiopankki.fi), Biobank Borealis of Northern Finland (www.ppshp.fi/Tutkimus-ja-opetus/Biopankki/Pages/Biobank-Borealis-briefly-in-English.aspx), Finnish Clinical Biobank Tampere (www.tays.fi/en-US/Research_and_development/Finnish_Clinical_Biobank_Tampere), Biobank of Eastern Finland (www.ita-suomenbiopankki.fi/en), Central Finland Biobank (www.ksshp.fi/fi-FI/Potilaalle/Biopankki), Finnish Red Cross Blood Service Biobank (www.veripalvelu.fi/verenluovutus/biopankkitoiminta), Terveystalo Biobank (www.terveystalo.com/fi/Yritystietoa/Terveystalo-Biopankki/Biopankki/) and Arctic Biobank (www.oulu.fi/en/university/faculties-and-units/faculty-medicine/northern-finland-birth-cohorts-and-arctic-biobank). All Finnish Biobanks are members of the BBMRI.FI infrastructure (www.bbmri-eric.eu/national-nodes/finland/). The Finnish Biobank Cooperative (FINBB) (https://finbb.fi/) is the coordinator of BBMRI-ERIC operations in Finland. The Finnish biobank data can be accessed through the Fingenious services (https://site.fingenious.fi/en/) managed by the FINBB. Supplementary Table 17 lists the members of FinnGen.

## Author contributions

J.R.S. and Y.L. coordinated the project. The core analytical team consisted of J.R.S., J.A.P., S.K., A.J. and J.N. M.T., K.L., A.B. and Y.L. proposed and designed the study. The core writing group consisted of J.R.S., J.A.P., S.K., A.J., J.N., M.T., K.L., A.B. and Y.L. J.R.S., J.A.P., S.K., A.J., J.N., A.H., E.H., K.K., N.F., R.Z., J.J.M., J.R.G., J.B., Y.X., E.K., B.C.A. and O.F. contributed to the data analysis or interpretation of the results. S.H., M.L., C.R., J.W., A.S., I.H., T.R.-K., T.M.W., P.F.S., O.A.A., M.T., K.L., A.B. and Y.L. contributed to the data or obtained funding. T.M.W., P.F.S., O.A.A., M.T., K.L., A.B. and Y.L. helped to conceive the study and supervised the project. All authors commented on the manuscript and approved the final version.

## Funding

## Competing interests

P.F.S. was a paid adviser and shareholder for Neumora Therapeutics. O.A.A. has received speaker fees from Lundbeck, Janssen, Otsuka and Sunovion, and is a consultant to Cortechs.ai and Precision Health. I.H. has received speaker honoraria from Lundbeck and Otsuka. The other authors declare no competing interests.

## Additional information

**Correspondence and requests for materials** should be addressed to John R. Shorter or Yi Lu.

# Reporting Summary

## Statistics

For all statistical analyses, confirm that the following items are present in the figure legend, table legend, main text, or Methods section.

| n/a | Confirmed | |
|---|---|---|
| ☐ | ☒ | The exact sample size (*n*) for each experimental group/condition, given as a discrete number and unit of measurement |
| ☐ | ☒ | A statement on whether measurements were taken from distinct samples or whether the same sample was measured repeatedly |
| ☐ | ☒ | The statistical test(s) used AND whether they are one- or two-sided<br>*Only common tests should be described solely by name; describe more complex techniques in the Methods section.* |
| ☐ | ☒ | A description of all covariates tested |
| ☐ | ☒ | A description of any assumptions or corrections, such as tests of normality and adjustment for multiple comparisons |
| ☐ | ☒ | A full description of the statistical parameters including central tendency (e.g. means) or other basic estimates (e.g. regression coefficient) AND variation (e.g. standard deviation) or associated estimates of uncertainty (e.g. confidence intervals) |
| ☐ | ☒ | For null hypothesis testing, the test statistic (e.g. *F*, *t*, *r*) with confidence intervals, effect sizes, degrees of freedom and *P* value noted<br>*Give P values as exact values whenever suitable.* |
| ☒ | ☐ | For Bayesian analysis, information on the choice of priors and Markov chain Monte Carlo settings |
| ☐ | ☒ | For hierarchical and complex designs, identification of the appropriate level for tests and full reporting of outcomes |
| ☐ | ☒ | Estimates of effect sizes (e.g. Cohen's *d*, Pearson's *r*), indicating how they were calculated |

*Our web collection on statistics for biologists contains articles on many of the points above.*

## Software and code

Policy information about availability of computer code

| Data collection | This study does not involve new data collection. |
|---|---|
| Data analysis | To ensure transparent and reproducible analyses of data across sites, we developed software containers and accompanying codes for data processing and analyses (e.g., GWAS, PRS, LD score regression), and distributed the containers across the study sites. All source codes and files are publicly available via GitHub. The central code repository for the tools for GWAS and post-GWAS analyses is https://github.com/comorment/containers, while other tools and public reference data are available in the same GitHub organization. |

For manuscripts utilizing custom algorithms or software that are central to the research but not yet described in published literature, software must be made available to editors and reviewers. We strongly encourage code deposition in a community repository (e.g. GitHub). See the Nature Portfolio guidelines for submitting code & software for further information.

## Data

Policy information about availability of data

All manuscripts must include a data availability statement. This statement should provide the following information, where applicable:
- Accession codes, unique identifiers, or web links for publicly available datasets
- A description of any restrictions on data availability
- For clinical datasets or third party data, please ensure that the statement adheres to our policy

> We have included a data availability statement as follows:
> The GWAS summary statistics reported in the study are made available through Figshare: https://doi.org/10.6084/m9.figshare.27830340.

## Research involving human participants, their data, or biological material

Policy information about studies with human participants or human data. See also policy information about sex, gender (identity/presentation), and sexual orientation and race, ethnicity and racism.

| | |
|---|---|
| Reporting on sex and gender | We used biological sex in the study. It was determined based on the participants' genotypes. |
| Reporting on race, ethnicity, or other socially relevant groupings | We did not consider race/ethnicity in this study.<br>We determined genetic ancestry of study participants within each cohort (details are provided in Supplementary Methods).<br>We performed GWAS meta-analyses combining the Nordic cohorts of individuals with European ancestry, since we had too few samples of other ancestries to warrant separate analyses. |
| Population characteristics | Details are provided for each cohort in the Supplementary Methods. |
| Recruitment | We provide detailed descriptions of the cohorts included in this study in the Supplementary Methods. |
| Ethics oversight | This study was approved by the Swedish ethical review authority (Dnr 2023-03073), and each cohort was approved by the relevant institutional review boards. |

Note that full information on the approval of the study protocol must also be provided in the manuscript.

# Field-specific reporting

Please select the one below that is the best fit for your research. If you are not sure, read the appropriate sections before making your selection.

☒ Life sciences    ☐ Behavioural & social sciences    ☐ Ecological, evolutionary & environmental sciences

For a reference copy of the document with all sections, see nature.com/documents/nr-reporting-summary-flat.pdf

# Life sciences study design

All studies must disclose on these points even when the disclosure is negative.

| | |
|---|---|
| Sample size | Following harmonization of phenotypic definitions of MDD and AAO across nine cohorts from five Nordic countries (Denmark, Estonia, Finland, Norway, and Sweden) (Methods), we identified a total of 151,582 MDD cases, including 46,708 eoMDD cases with age of first diagnosis ≤ 25 years old (approximating an AAO ≤ 20-21; Methods) and 37,168 loMDD cases with age of first diagnosis ≥ 50 (approximating an AAO ≥ 44-45; Methods) (Table 1). |
| Data exclusions | n/a |
| Replication | To assess generalizability outside the Nordic cohorts, we also analyzed the UK Biobank data which relied on the self-reported age at first diagnosis, and conducted the GWAS of eoMDD and loMDD with the same age cutoffs (Fig. S1). However, considering the major differences in samples and phenotypes, we conducted primary analyses based on Nordic samples, with the UK Biobank as a replication cohort for the identified loci. Top loci were partly replicated in the UK Biobank, with one locus on chromosome 9 showing nominal significance (P<0.05); however, for the genome-wide significant loci in either sample, the correlation in their effect sizes was substantial (r=0.84 for eoMDD. Fig. S3, Tables S3-S5). |
| Randomization | This was a genetic association study. Allocation by genotype. |
| Blinding | This was a genetic association study, following observational design. So no blinding was used. |

# Reporting for specific materials, systems and methods

We require information from authors about some types of materials, experimental systems and methods used in many studies. Here, indicate whether each material, system or method listed is relevant to your study. If you are not sure if a list item applies to your research, read the appropriate section before selecting a response.

## Materials & experimental systems

| n/a | Involved in the study |
|-----|----------------------|
| ☒ ☐ | Antibodies |
| ☒ ☐ | Eukaryotic cell lines |
| ☒ ☐ | Palaeontology and archaeology |
| ☒ ☐ | Animals and other organisms |
| ☒ ☐ | Clinical data |
| ☒ ☐ | Dual use research of concern |
| ☒ ☐ | Plants |

## Methods

| n/a | Involved in the study |
|-----|----------------------|
| ☒ ☐ | ChIP-seq |
| ☒ ☐ | Flow cytometry |
| ☒ ☐ | MRI-based neuroimaging |

## Plants

| | |
|--|--|
| Seed stocks | N/A |
| Novel plant genotypes | N/A |
| Authentication | N/A |

