## [Peer Review File · Nature Genetics]

Genome-wide association analyses identify distinct genetic architectures for early- and late-onset depression

Corresponding Author: Dr John Shorter

Version 0:

Decision Letter:

26th January 2025

Dear Dr. Shorter,

Your Letter "Genome-wide association study using Nordic biobank and longitudinal health registry reveals different genetic architecture between early- and late-onset depression" has been seen by three referees. You will see from their comments below that, while they find your work of interest, they have raised several relevant points. We are interested in the possibility of publishing your study in Nature Genetics, but we would like to consider your response to these points in the form of a revised manuscript before we make a final decision on publication.

To guide the scope of the revisions, the editors discuss the referee reports in detail within the team, including with the chief editor, with a view to identifying key priorities that should be addressed in revision, and sometimes overruling referee requests that are deemed beyond the scope of the current study. In this case, we ask that you clarify and justify the cutoffs used to define early-onset and late-onset cases, perform further analyses to assess heterogeneity across study cohorts, address other technical queries with suitable revisions and clarifications, and extend the analyses where feasible as requested by the referees. We hope you will find this prioritized set of referee points to be useful when revising your study. Please do not hesitate to get in touch if you would like to discuss these issues further.

We therefore invite you to revise your manuscript taking into account all reviewer and editor comments. Please highlight all changes in the manuscript text file. At this stage, we will need you to upload a copy of the manuscript in MS Word .docx or similar editable format.

*2) If you have not done so already, please begin to revise your manuscript so that it conforms to our Letter format instructions, available

[here](http://www.nature.com/ng/authors/article_types/index.html).

*3) Include a revised version of any required Reporting Summary: <https://www.nature.com/documents/nr-reporting-summary.pdf>

EXTENDED DATA FIGURES

Link Redacted

We hope to receive your revised manuscript within 8-12 weeks. If you cannot send it within this time, please let us know.

Nature Genetics is committed to improving transparency in authorship. As part of our efforts in this direction, we are now requesting that all authors identified as 'corresponding author' on published papers create and link their Open Researcher and Contributor Identifier (ORCID) with their account on the Manuscript Tracking System (MTS), prior to acceptance. ORCID helps the scientific community achieve unambiguous attribution of all scholarly contributions. You can create and link your ORCID from the home page of the MTS by clicking on 'Modify my Springer Nature account'. For more information, please visit <http://www.springernature.com/orcid>.

Sincerely,
Kyle

Kyle Vogan, PhD
Senior Editor
Nature Genetics
<https://orcid.org/0000-0001-9565-9665>

Referee expertise:

Referee #1: Genetics, psychiatric disorders

Referee #2: Genetics, psychiatric disorders

Referee #3: Genetics, psychiatric disorders

Reviewers' Comments:

Reviewer #1 (Remarks to the Author):

General

This work uses data from the Nordic health registers to establish two age of onset defined subgroups of MDD: eoMDD and loMDD, and use a compendium of statistical genetics approaches to analyse the differences in their genetic architecture, functional genomics, and putative causal relationships and comorbidity with other phenotypes and diseases of interest. It is the first largest and comprehensive analysis of age of onset subtypes of MDD that holds promise to set industry standards going forward. I think the following should be done in order to truly establish this work as a basis for future work.

Major

1. Why was age of diagnosis ≤ 25 years chosen for eoMDD? The authors wrote previously "Previous attempts to stratify MDD by AAO have been hindered by methodological challenges, including large variations of AAO across samples, recall bias, and relatively small sample size" - it seems the authors are trying to resolve these issues through establishing some industry standards as to what early onset means. In this case I would need to see some justification how this decision of age cutoff was reached, ideally through a combination of literature review or empirical exploration of the data they have used. The authors wrote they "leveraged the Nordic biobanks and harmonized longitudinal health registries to stratify MDD cases based on age at first MDD diagnosis", which makes me think they derived this age cutoff based on empirical evidence of sorts (epidemiology? genetic correlations between eoMDD or loMDD?), and I think it

is essential to see this fully explained in the paper (supplementary methods ok) if the authors want to claim that this age cutoff is the one to be used in subsequent studies. The same goes for the age of diagnosis >50 for loMDD.

2. Figure 1 and Table 1 show that the study composition of eoMDD and loMDD cases are different (loMDD lacks data from iPSYCH and MoBA). Given that 1) the authors wrote later that they "observed high genetic correlations (0.7-0.9) among the largest Nordic cohorts (iPSYCH, ESTBB, FinnGen)", 2) iPSYCH is one of the cohorts with high rG and contribute the largest sample size to eoMDD but it is not present in loMDD, and 3) PREFECT making up some of the loMDD seems to have really low rG with other cohorts in MDD and eoMDD (why is its rG with EST and FINN not even shown in Fig S1?), I have reasons to believe that the low rG (0.58) between eoMDD and loMDD is at least partially attributable to cohort differences rather than age of onset differences. Have the authors performed within cohort rG analysis between loMDD and eoMDD (in EST and FINN for example)? It would be good to show how those results vary from the results using all cohorts.

3. Chromatin marks analysis: how much of the difference in significance of this analysis between eoMDD and loMDD can be attributable to sample size and therefore power or cohort differences between the two (eoMDD has 10K more cases)? It seems across the board there are fewer chromatin marks associated with loMDD than eoMDD, suggesting this may be a power issue or a cohort effect issue - I would recommend the authors use downsampling or leave-one-cohort-out analysis to test robustness of these findings.

Minor

4. This should probably be more precisely explained - "After controlling for loMDD, the genetic associations of eoMDD with other traits remained similar, except that the negative rg between eoMDD and educational attainment was attenuated (Table S9)." So, since this is a GenomicSEM analysis, it would be regressing out genetic effects of loMDD out of the phenotypes assessed?

5. MR results: are all results reported in the paper significant? I can't find a significance threshold or statistics in either the plot, main text or methods, please state to let readers know how the multiple-testing correction is done and how significance is determined.

Reviewer #2 (Remarks to the Author):

The authors present a study using Nordic biobank data to perform genome-wide association analyses of early onset depression and late onset depression, compare the results, and carry out additional analyses to explore the association with other phenotypes. They sample size is over 46k early onset (<25 yrs) cases and over 37k late onset (>50 yrs) cases. They observed 12 hits for early onset and 2 hits for late onset depression. Early onset was significantly more associated with several phenotypes including suicide attempts. Following up on that result, they report that polygenic risk for eoMDD was highly associated with risk of suicide attempt in the ten years after MDD diagnosis.

This study is performed in a reasonably large sample and the methods are sound. I have some questions and comments:

- Minor comment: The meta-analysis by Solmi et al. published in Molecular Psychiatry on age at onset of mental disorders showed that peak age of onset for depression is 19 years and over a third of people have the onset before 25 years. I think that's worth mentioning in the introduction to take away the impression that an onset below 25 years of age is exceptional.

- Is there a reason the authors have restricted their analyses to the Nordic countries as there are several longitudinal cohorts that have collected data on depression. In the PGC depression cohorts, there may also be information on age at onset, which could increase sample sizes.

- The authors state that they haven't done Mendelian randomisation analyses for other psychiatric disorders as "these relationships likely reflect shared biological etiology". I think it would be interesting to do MR analyses with the childhood onset diagnoses ASD and ADHD. These diagnoses are often comorbid with depression at a later stage and the question whether they cause depression is still unanswered.

- I think they overinterpret the results of the difference between the associations with CVD and eoMDD and loMDD. The genetic correlation is only significantly different for heart failure and the correlation between stroke and loMDD is decreased when controlling for eoMDD. These effects are not seen for CVD and peripheral artery disease.

- Minor comment, in line 200 they refer to "comorbid diagnoses of bipolar disorder and schizophrenia". Especially for bipolar disorder, I'd think this is a subsequent diagnosis replacing the depression diagnosis, not so much a comorbid diagnosis.

- I find the follow up analyses to explore the potential usefulness of the early onset depression PRS as a predictor for suicide attempt interesting. I wonder why the authors have chosen to restrict these analyses to individuals diagnosed with depression. I think it would be interesting to explore this for other psychiatric phenotypes as well as most of them, including ADHD and ASD, are also related to suicide attempts.

Reviewer #3 (Remarks to the Author):

The study examines the genetic architecture of major depressive disorder utilising genome-wide genotype data from Nordic biobanks and electronic health registry data. Age of onset is used to reduce heterogeneity and the genetic architecture of early onset MDD (eoMDD) and late onset MDD (loMDD) compared. Twelve genomic loci for eoMDD and two loci for loMDD were identified. Evidence that eoMDD is related to early brain development and is less polygenic than loMDD was presented.

MDD is a major issue for humanity. There is great need for better prevention and treatment strategies. It is clear that the heterogeneity of MDD is a barrier to effective invention and more personalised/stratified approaches are required. The study is of large scale and the longitudinal health record data enables more reliable testing of prediction models than is possible with cross sectional samples (which are common in the field). The hypothesis that early and late onset forms of MDD have a different genetic architecture has face validity. Evidence for the use of age of diagnosis as a proxy for age of onset is provided and seems entirely reasonable. For meaningful meta-analysis process of phenotype data harmonisation is key. The authors have made a substantial effort to harmonise the data. State of the art GWAS and post GWAS analysis has been applied. I cannot see any technical issues with the analysis.

I have a few queries which I hope the authors may be able to answer:

- 1) I understood that the 25th and 75th centile boundaries of the MDD age of onset distribution were used as the thresholds for eoMDD and loMDD. Was there any other rationale for using the age thresholds?
- 2) For eoMDD cases, was there a minimum length of time from diagnosis for participant to be included in the study? I suppose I am slightly concerned about conversion to a bipolar disorder diagnosis.
- 3) Was GWAS with age as co-variate performed for comparison?

I also have a few comments which the authors may like to address:

- 1) I appreciate that only the significant loci were tested in the UK Biobank but given the relatively small number of cases in the UK Biobank compared with discovery sample, I am not convinced that it can be considered a replication sample. I think it might be more helpful to present the direction of association of the risk allele for each constituent sample.
- 2) Given debate about relationship between depression and Alzheimer's disease, it might be of interest to highlight the negative finding for correlation.
- 3) A discussion of the AAO finding in the context of other clinical characteristic which may reduce heterogeneity would, I think, be of value.
- 4) The paper emphasises the potential clinical utility of genetic subtyping of MDD by AAO. While I am strong advocate of such an approach, I fear that the utility is somewhat overstated. In particular, the difference between the 10-year absolute risk for people in lowest and highest PRS deciles (as presented in abstract) appears impressive. However, the difference between the highest decile and the "middle" 8 deciles is small. In any case, the risk of suicide attempt is so high across the board I cannot see that PRS calculated risk would change clinical management. Furthermore, the result of the MR analysis indicating that eoMDD is likely causative of adverse cardiovascular outcomes is presented in terms of its clinical utility. While this finding is definitely of pathophysiological and clinical interest, I think the practical implications are limited.

Version 1:

Decision Letter:

Our ref: NG-LE67100R

12th June 2025

Dear John,

Your revised manuscript "Genome-wide association study using Nordic biobank and longitudinal health registry reveals different genetic architecture between early- and late-onset depression" (NG-LE67100R) has been seen by the original referees. As you will see from their comments below, they find that the paper has improved in revision, and therefore we will be happy in principle to publish it in Nature Genetics as a Letter pending final revisions to comply with our editorial and formatting guidelines.

We are now performing detailed checks on your paper, and we will send you a checklist detailing our editorial and formatting requirements soon. Please do not upload the final materials or make any revisions until you receive this additional information from us.

Thank you again for your interest in Nature Genetics. Please do not hesitate to contact me if you have any questions.

Sincerely,
Kyle

Kyle Vogan, PhD
Senior Editor
Nature Genetics
<https://orcid.org/0000-0001-9565-9665>

Reviewer #1 (Remarks to the Author):

The authors have addressed my questions.

Reviewer #2 (Remarks to the Author):

The authors have incorporated most of the comments. They have also explained why they think additional analyses focusing on ADHD and ASD (comment R2.3 and R2.6) are not within the scope of this paper. Still, I don't agree. I think given the high comorbidity between ADHD, ASD and early onset depression it is timely to explore the genetic associations with ADHD and ASD (and with suicidal attempts) and not leave it to other to follow up on the results in this paper. However, if the editor also feels it is too much for one paper, it's okay for me if the analyses are not included.

Reviewer #3 (Remarks to the Author):

I thank the authors for comprehensively addressing all the comments and queries I raised. I have no further comments.

Response to Reviewers – NG-LE67100 “Genome-wide association study using Nordic biobank and longitudinal health registry reveals different genetic architecture between early- and late-onset depression”

We would like to express our gratitude to the editor and reviewers for constructive suggestions. We have carefully considered your recommendations and have responded to each of them below. We truly appreciate the time and effort you invested in reviewing our work, and believe that the revisions made because of your feedback have strengthened our study. Changes are outlined below and highlighted in the revised manuscript NG-LE67100_R1_main_TrackChanges.docx, and a clean manuscript (NG-LE67100_R1_main_cleaned.docx).

Editorial comments:

We ask that you clarify and justify the cutoffs used to define early-onset and late-onset cases, perform further analyses to assess heterogeneity across study cohorts, address other technical queries with suitable revisions and clarifications, and extend the analyses where feasible as requested by the referees. We hope you will find this prioritized set of referee points to be useful when revising your study. Please do not hesitate to get in touch if you would like to discuss these issues further.

Thank you for the guidance. We have implemented revisions in response to the prioritized set of referees' points:

- 1. Clarified and justified the cutoffs for early- and late-onset cases (see responses to R1.1, R2.1, R3.1)*
- 2. Performed additional analyses to assess heterogeneity across study cohorts (R1.2).*
- 3. Extended analyses where feasible as requested by the referees: e.g., R1.3 to investigate the impact of sample size on chromatin mark analyses, R2.3 to investigate the potential causal effect of neurodevelopmental disorders on MDD.*
- 4. Addressed the technical queries regarding UKB analyses, GenomicSEM, MR, etc.*

Reviewers' Comments:

Reviewer #1:

General. This work uses data from the Nordic health registers to establish two age of onset defined subgroups of MDD: eoMDD and loMDD, and use a compendium of statistical genetics approaches to analyse the differences in their genetic architecture, functional genomics, and putative causal relationships and comorbidity with other phenotypes and diseases of interest. It is the first largest and comprehensive analysis of age of onset subtypes of MDD that holds promise to set industry standards going forward. I think the following should be done in order to truly establish this work as a basis for future work.

Major. **R1.1.** Why was age of diagnosis ≤ 25 years chosen for eoMDD? The authors wrote previously "Previous attempts to stratify MDD by AAO have been hindered by methodological challenges, including large variations of AAO across samples, recall bias, and relatively small sample size" - it seems the authors are trying to resolve these issues through establishing some industry standards as to what early onset means. In this case I would need to see some justification how this decision of age cutoff was reached, ideally through a combination of literature review or empirical exploration of the data they have used. The authors wrote they "leveraged the Nordic biobanks and harmonized longitudinal health registries to stratify MDD cases based on age at first MDD diagnosis", which makes me think they derived this age cutoff based on empirical evidence of sorts (epidemiology? genetic correlations between eoMDD or loMDD?), and I think it is essential to see this fully explained in the paper (supplementary methods ok) if the authors wants to claim that this age cutoff is the one to be used in subsequent studies. The same goes for the age of diagnosis >50 for loMDD.

*We thank the reviewer for highlighting the need to justify the cutoffs. Age at onset or first diagnosis is inherently continuous, so no single “gold-standard” threshold exists to define early- and late-onset MDD (the cutoffs vary in the literature, **Table 1**). Our intention was therefore to define epidemiologically sound subgroups with sufficient sample sizes to assess if any biologically meaningful difference between them exists.*

*We selected age at first diagnosis ≤ 25 years for eoMDD and ≥ 50 years for loMDD based on literature review (particularly, Solmi et al. meta-analysis¹; **Table 1**) and our empirical population registry data (**Table 1, Fig. 1**).*

As presented in our Online Methods, Solmi et al. (2022) meta-analysis showed: a) the median AAO for MDD is around age 30, with the 25th and 75th percentile at age 21 and 44, respectively; b) on average, the onset predates first diagnosis by 5 years¹. We thus considered individuals with their first psychiatric treatment contact for MDD

≤age 25 (approximate to AAO p25 at age 20-21) as eoMDD, and those with first psychiatric contact ≥age 50 (approximate to AAO p75 at age 44-45) as loMDD. These cutoffs were also commonly used in classic literature^{2,3}.

Table 1. Summary of key studies of MDD age-at-onset from >4,000 samples.

First author (Year)	PMID	Cohort	Measure	phenotype ^a	mean/median, p25&75	Cutoffs for grouping
Zisook (2007)	17898345	STAR*D	self-report	AAO	26	5 AAO groups
Fernandez-Pujals (2015)	26571028	GenScotland	self-report	AAO	31.7 35%: age 25	earlier/later onset with cutoff age 40
Power (2017)	27519822	multiple cohorts at the PGC	self-report	AAO	27	Cutoffs vary widely across cohorts
Solmi (2022)	34079068	meta of 62 studies	self-report	either	30 p25: age 21 p75: age 44	–
Solmi (2022)	34079068	meta of 22 studies	self-report	AAO	26 p25: age 17 p75: age 34	–
Solmi (2022)	34079068	meta of 35 studies	self-report	AAD	31 p25: age 21 p75: age 46	–
Harder (2022)	35347114	UK Biobank	self-report	AAO	37.3	–
Harder (2022)	35347114	UK Biobank	self-report	AAD	42.7	–
unpublished	–	Swedish birth cohort 1958-1993	register	AAD ^b	33.0 p25: age 25 p75: age 45	–

a. Phenotype of age-at-onset (AAO), or age-at-first diagnosis (AAD) as proxy. b. Age at first specialist/psychiatrist diagnosis in Swedish data. Abbreviations: p25, p75: 25th and 75th percentile. PGC: Psychiatric Genomics Consortium.

The cutoffs were further supported by empirical data from our whole-population health registry. We used data from Sweden as an example, but our previous research demonstrated that the distributions of age-at-first-diagnosis were similar in Denmark⁵. First, we observed a similar distribution of age-at-first-diagnosis in the Swedish birth cohort 1958-1993 as reported in Solmi et al (Table 1). Additionally, our cutoffs were selected to capture the peaks of MDD incidence in the population⁴ (Fig. 1) while balancing with sufficient sample sizes from the biobanks.

In summary, our literature review and empirical data justify the chosen cutoffs, and our genetic findings provide further evidence that these cutoffs indeed reflect biological differences between eoMDD and loMDD. We have revised Methods to include the justification of the cutoff (line 391-399).

Fig. 1. Standardized incidence rate per 1000 person years (y-axis) by age (x-axis) for depressive disorders in Swedish register 2003-2019 (Fig 1b in Yang et al. 2024⁴). Red: female; Blue: male.

R1.2 Figure 1 and Table 1 show that the study composition of eoMDD and loMDD cases are different (loMDD lacks data from iPSYCH and MoBA). Given that 1) the authors wrote later that they "observed high genetic correlations (0.7-0.9) among the largest Nordic cohorts (iPSYCH, ESTBB, FinnGen)", 2) iPSYCH is one of the cohorts with high rG and contribute the largest sample size to eoMDD but it is not present in loMDD, and 3) PREFECT making up some of the loMDD seems to have really low rG with other cohorts in MDD and eoMDD (why is its rG with EST and FINN not even shown in Fig S1?), I have reasons to believe that the low rG (0.58)

between eoMDD and loMDD is at least partially attributable to cohort differences rather than age of onset differences. Have the authors performed within cohort rG analysis between loMDD and eoMDD (in EST and FINN for example)? It would be good to show how those results vary from the results using all cohorts.

We thank the reviewer for raising this point. In addition to our phenotype harmonization efforts to minimize cohort differences, we now performed two analyses, and the findings did not suggest strong cohort differences.

First, the within-cohort rG between eoMDD and loMDD was 0.45 and 0.73 for EstBB and FinnGen, respectively (UKB: 0.40), and the cross-cohort rG ranged between 0.37-0.65 (Fig. 2A. Added to main text as Fig. S1D). The

rG based on the meta-analyses of eoMDD and loMDD (0.58) was within these ranges. Second, the leave-one-out PRS associations were largely consistent across major cohorts such as iPSYCH, EstBB, FinnGen (Fig. 2B).

Fig. 2. (A). *rG* in largest cohorts with both eoMDD and loMDD phenotypes (EstBB, FinnGen, and UKB). * indicating *rG* was significant after correction for multiple testing. (B). Leave-one-out PRS associations with MDD phenotypes across cohorts. Red, blue, and yellow indicate PRS of eoMDD, loMDD, and broad MDD, respectively.

R1.3 Chromatin marks analysis: how much of the difference in significance of this analysis between eoMDD and loMDD can be attributable to sample size and therefore power or cohort differences between the two (eoMDD has 10K more cases)? It seems across the board there are fewer chromatin marks associated with loMDD than eoMDD, suggesting this may be a power issue or a cohort effect issue - I would recommend the authors use downsampling or leave-one-cohort-out analysis to test robustness of these findings.

The reviewer correctly noted that the differences observed in these analyses could reflect the larger sample size for eoMDD compared to loMDD. Since our analyses above (Fig. 2) suggested limited evidence for strong cohort differences, we performed leave-one-out (LOO) analyses, as suggested, to investigate this possibility. Specifically, we removed the EstBB ($N \approx 9,000$) or FinnGen ($N \approx 10,000$) cohorts from the eoMDD GWAS meta, making the eoMDD and loMDD sample sizes roughly comparable. The results of these LOO analyses were largely consistent with the full eoMDD meta-analysis (Fig. 3), suggesting that the observed differences are unlikely to be driven by sample size disparities between the two subgroups.

Fig. 3 Enrichment of open chromatin marks in eoMDD and loMDD. eoMDD GWAS after leaving EstBB and FinnGen out.

Minor. **R1.4** This should probably be more precisely explained - "After controlling for loMDD, the genetic associations of eoMDD with other traits remained similar, except that the negative *rG* between eoMDD and

educational attainment was attenuated (Table S9)." So, since this is a GenomicSEM analysis, it would be regressing out genetic effects of lOMDD out of the phenotypes assessed?

The reviewer is correct, as stated in the Methods (line 654): "Next, we assessed the subtype-specific contribution to the other phenotypes by controlling for the other subtype (e.g., we investigated the relationship between eoMDD and suicide attempt after controlling for lOMDD). In this manner, the unique contribution of each can be gauged while controlling for their shared variance."

To further clarify this, we added the following sentence to the Results (line 155-157): "In this way, the unique overlap between each subtype and the other traits could be gauged, giving insight into how much of the overlap with the other trait was driven by overlap with the other subtype."

R1.5. MR results: are all results reported in the paper significant? I can't find a significance threshold or statistics in either the plot, main text or methods, please state to let readers know how the multiple-testing correction is done and how significance is determined.

We have now added an FDR correction, with significance indicated in the figure. Please note that the emphasis is on effect consistency across sensitivity analyses, rather than any single p-value, as has now been explained in the Methods (line 671-673): "For interpretation, we focus on effect consistency across tests, rather than p-values. Still, to give an indication of the extent of the multiple testing burden, we report significance according to FDR-corrected p-values (Benjamini-Hochberg method)."

Reviewer #2:

The authors present a study using Nordic biobank data to perform genome-wide association analyses of early onset depression and late onset depression, compare the results, and carry out additional analyses to explore the association with other phenotypes. They sample size is over 46k early onset (<25 yrs) cases and over 37k late onset (>50 yrs) cases. They observed 12 hits for early onset and 2 hits for late onset depression. Early onset was significantly more associated with several phenotypes including suicide attempts. Following up on that result, they report that polygenic risk for eoMDD was highly associated with risk of suicide attempt in the ten years after MDD diagnosis.

This study is performed in a reasonably large sample and the methods are sound. I have some questions and comments:

R2.1 Minor comment: The meta-analysis by Solmi et al. published in Molecular Psychiatry on age at onset of mental disorders showed that peak age of onset for depression is 19 years and over a third of people have the onset before 25 years. I think that's worth mentioning in the introduction to take away the impression that an onset below 25 years of age is exceptional.

We appreciate the reviewer's suggestions. As noted in our response R1.1, Solmi et al. meta-analysis served as our primary reference for determining the cutoffs. We have now clarified in the Introduction that the cutoffs of ≤ 25 years and ≥ 50 years align with the first and third quartile of the AAO distribution, respectively (line 87-88).

R2.2 Is there a reason the authors have restricted their analyses to the Nordic countries as there are several longitudinal cohorts that have collected data on depression. In the PGC depression cohorts, there may also be information on age at onset, which could increase sample sizes.

*The reviewer is correct that other data sources, such as longitudinal cohorts and PGC-MDD cohorts, are available. We note that a previous PGC-MDD effort (Power et al. 2017, **Table 1**) highlighted the substantial heterogeneity in AAO phenotypes, reflecting study-specific definitions (e.g., age at first symptoms, first visit to general practitioner, or first diagnosis). Additionally, there is an ongoing effort aiming to define adolescent depression across several prospective and retrospective cohorts, including the PGC-MDD with limited number of cases (Grimes et al., personal communication).*

In contrast, our approach strikes a practical balance between phenotype harmonization and sample size. We have made considerable efforts to harmonize the phenotypes (diagnostic records from healthcare registry data) across the Nordic countries with similar healthcare systems⁵, while assembling a sufficiently large N to maintain robust statistical power for downstream analyses.

R2.3 The authors state that they haven't done Mendelian randomisation analyses for other psychiatric disorders as "these relationships likely reflect shared biological etiology". I think it would be interesting to do MR analyses

with the childhood onset diagnoses ASD and ADHD. These diagnoses are often comorbid with depression at a later stage and the question whether they cause depression is still unanswered.

We thank the reviewer for this interesting suggestion. We have conducted a preliminary analysis of the effect of ADHD and ASD on MDD (given the childhood-onset of these disorders, we only tested the effect in this direction) (Fig. 4). We did not observe significant pleiotropy according to the MR-Egger estimate. However, Steiger filtering showed that for ASD, half of the instruments explained more variance in MDD than in ASD, leaving only 5 unidirectional instruments for analysis.

Even though these preliminary results suggest that these analyses are statistically feasible, we decide not to present them in the manuscript. When applying MR, there should be a clear hypothesis on the causal association tested, which is more difficult for conditions that often share symptoms. This type of analysis would fit better in a paper where more emphasis can be devoted to theorizing, identifying non-pleiotropic instruments, and triangulation. We refer the reviewer to a recent paper that used MR among other epidemiological methods to test the causal association between ADHD and MDD⁶. For ASD, such a study is still lacking, which is likely because its GWAS still lacks power to yield reliable MR instruments.

Fig. 4 Causal relationship between ASD, ADHD, MDD phenotypes.

R2.4 I think they overinterpret the results of the difference between the associations with CVD and eoMDD and loMDD. The genetic correlation is only significantly different for heart failure and the correlation between stroke and loMDD is decreased when controlling for eoMDD. These effects are not seen for CVD and peripheral artery disease.

Thank you for raising caution about potential overinterpretation. We have now revised the paragraph starting from line 265 to provide a more balanced interpretation of the CVD results:

“The genetic associations between eoMDD, loMDD, and somatic comorbidities confirm the role of shared genetics in these comorbidities. However, we also detected some differences between eoMDD and loMDD, suggesting variability in genetic links to comorbid conditions. Previous studies have indicated that MDD increases the risk for CVD and that genetics play a role in the MDD-CVD comorbidity³⁰. Our Genomic SEM and PheWAS analyses reveal that the associations with heart failure and stroke may be primarily driven by the eoMDD subtype, emphasizing the importance of MDD subtyping in the clinical management of MDD patients.”

R2.5 Minor comment, in line 200 they refer to "comorbid diagnoses of bipolar disorder and schizophrenia". Especially for bipolar disorder, I'd think this is a subsequent diagnosis replacing the depression diagnosis, not so much a comorbid diagnosis.

We have now revised the phrase as “diagnostic conversion to bipolar disorder and schizophrenia” (line 209).

R2.6 I find the follow up analyses to explore the potential usefulness of the early onset depression PRS as a predictor for suicide attempt interesting. I wonder why the authors have chosen to restrict these analyses to individuals diagnosed with depression. I think it would be interesting to explore this for other psychiatric phenotypes as well as most of them, including ADHD and ASD, are also related to suicide attempts.

Thank you for your insightful comments. We restricted our analyses to individuals diagnosed with MDD to enhance the relevance of our findings within the context of MDD. We acknowledge the potential value of exploring the predictive utility of the eoMDD PRS in other suicide related psychiatric conditions such as ADHD and ASD. While a broader scope may indeed yield interesting findings, these warrant a separate investigation with careful considerations of potential clinical applications. Our current focus on MDD aims to build groundworks for more targeted interventions to improve clinical outcomes for individuals with MDD.

Reviewer #3:

The study examines the genetic architecture of major depressive disorder utilising genome-wide genotype data from Nordic biobanks and electronic health registry data. Age of onset is used to reduce heterogeneity and the genetic architecture of early onset MDD (eoMDD) and late onset MDD (loMDD) compared. Twelve genomic loci for eoMDD and two loci for loMDD were identified. Evidence that eoMDD is related to early brain development and is less polygenic than loMDD was presented.

MDD is a major issue for humanity. There is great need for better prevention and treatment strategies. It is clear that the heterogeneity of MDD is a barrier to effective invention and more personalised/stratified approaches are required. The study is of large scale and the longitudinal health record data enables more reliable testing of prediction models than is possible with cross sectional samples (which are common in the field). The hypothesis that early and late onset forms of MDD have a different genetic architecture has face validity. Evidence for the use of age of diagnosis as a proxy for age of onset is provided and seems entirely reasonable. For meaningful meta-analysis process of phenotype data harmonisation is key. The authors have made a substantial effort to harmonise the data. State of the art GWAS and post GWAS analysis has been applied. I cannot see any technical issues with the analysis.

I have a few queries which I hope the authors may be able to answer:

R3.1 I understood that the 25th and 75th centile boundaries of the MDD age of onset distribution were used as the thresholds for eoMDD and loMDD. Was there any other rationale for using the age thresholds?

We appreciate the reviewer's positive evaluation and constructive comments, especially regarding the broader context. For this question, we would like to refer the reviewer to our response to R1.1 where we have outlined the epidemiological and empirical justification of the age thresholds.

R3.2 For eoMDD cases, was there a minimum length of time from diagnosis for participant to be included in the study? I suppose I am slightly concerned about conversion to a bipolar disorder diagnosis.

Thank you for this comment. We did not apply a unified minimum follow-up restriction across participating cohorts. However, our registers with extensive longitudinal records (except for the youngest) allowed the exclusion of participants with lifetime diagnosis of bipolar disorder or schizophrenia prior or after MDD diagnosis (see our exclusion criteria in Methods, line 386).

For eoMDD, the cohorts contributing most young cases were iPSYCH2012 and iPSYCH2015. The iPSYCH samples comprised individuals born between 1981 and 2008, with MDD/BD/SCZ diagnoses identified through December 31, 2015⁷. This may indeed be a limitation for the youngest subjects, which we added in line 316-318.

R3.3 Was GWAS with age as co-variate performed for comparison?

Yes, this is correct, and we explain our GWAS in more detail in the Methods, starting at line 578.

I also have a few comments which the authors may like to address:

R3.4 I appreciate that only the significant loci were tested in the UK Biobank but given the relatively small number of cases in the UK Biobank compared with discovery sample, I am not convinced that it can be considered a replication sample. I think it might be more helpful to present the direction of association of the risk allele for each constituent sample.

We agree with the reviewer that the UK-Biobank analyses do not constitute a replication due to the sample and phenotype differences. We now refer to it as our "comparison cohort" throughout the text.

Despite the differences, we anticipated some concordance in the SNP effects and effect directions. We added the effect direction for each constituent sample in the table showing the genome-wide significant hits for early-onset, late-onset, and broad MDD (Table S1). We also plotted the effect sizes of the independent GWAS loci in the Nordic meta-analysis versus the UK-Biobank GWAS (Fig. 5, next page). The top loci were mostly in the same directions, with small differences in effect size (and the meta-analysis generally having stronger effect sizes due to power). Of all significant loci (not limiting to the top ones), very few were in opposite directions: 0.6% for early-onset MDD, 6.8% for late-onset MDD, and 4.5% for all MDD.

R3.5 Given debate about relationship between depression and Alzheimer’s disease, it might be of interest to highlight the negative finding for correlation.

We thank the reviewer for bringing up the interesting negative finding of the genetic links between depression subtypes and Alzheimer’s disease. We have now included the following paragraph in the discussion (line 272-276).

“..., while the link between MDD and Alzheimer’s disease is well established^{8,9}, previous research did not identify strong shared genetic factors¹⁰. Similarly, we did not find significant genetic associations in either eoMDD or loMDD. While our subtype GWAS may have lower statistical power, the findings suggest that the magnitude of overall genetic associations between MDD and Alzheimer’s disease may be low.”

R3.6 A discussion of the AAO finding in the context of other clinical characteristic which may reduce heterogeneity would, I think, be of value.

We thank the reviewer for this suggestion. Indeed, previous studies, including our own, have shown that beyond AAO, other clinical characteristics such as vegetative symptoms, psychotic features, and disability also contribute to phenotypic and genetic heterogeneity in MDD^{11,12}. Our approach can be extended to these clinical characteristics to help reduce disorder heterogeneity. We have revised the last paragraph in Discussion:

“MDD genomics has achieved tremendous progress, from 0 significant findings in 2013 to 697 associations in 2025, for the MDD case-control phenotype. Here, we explore the strategy of going beyond case-control to target specific phenotypic subgroups, aiming to reduce genetic heterogeneity in MDD. Our harmonized healthcare data and analyses across countries improved analytical power and identified differential genetic signals in AAO-based subtypes. A similar approach can be extended to other clinical characteristics, such as vegetative symptoms, psychotic features, and disability, to further target clinically relevant subtypes of MDD. These findings may have important implications for guiding targeted treatment and prevention in psychiatry.”

R3.7 The paper emphasizes the potential clinical utility of genetic subtyping of MDD by AAO. While I am strong advocate of such an approach, I fear that the utility is somewhat overstated. In particular, the difference between the 10-year absolute risk for people in lowest and highest PRS deciles (as presented in abstract) appears impressive. However, the difference between the highest decile and the “middle” 8 deciles is small. In any case, the risk of suicide attempt is so high across the board I cannot see that PRS calculated risk would change clinical management. Furthermore, the result of the MR analysis indicating that eoMDD is likely causative of adverse cardiovascular outcomes is presented in terms of its clinical utility. While this finding is definitely of pathophysiological and clinical interest, I think the practical implications are limited.

We appreciate the reviewer’s insightful comments regarding the clinical utility of our findings. We fully acknowledge that the present results may not have immediate practical implications in mental health care.

Fig. 5 Effect comparisons of the GWA-significant SNPs in the UKB.

Rather, they primarily serve as a proof-of-principle, laying the groundwork for future investigations into both molecular mechanisms and predictive modeling.

To avoid overinterpretation of our findings, we revised the text by 1) adding the absolute risk in the middle group as a comparison (Abstract, Discussion), 2) reducing the emphases on the potential clinical utility of the PRS findings (Results, Discussions).

References

1. Solmi M, Radua J, Olivola M, et al. Age at onset of mental disorders worldwide: large-scale meta-analysis of 192 epidemiological studies. *Mol Psychiatry*. Jan 2022;27(1):281-295. doi:10.1038/s41380-021-01161-7
2. Parker G, Wilhelm K, Asghari A. Early onset depression: the relevance of anxiety. *Soc Psychiatry Psychiatr Epidemiol*. Jan 1997;32(1):30-7. doi:10.1007/BF00800665
3. H V, YV G. Late-onset depression: issues affecting clinical care. *Advances in Psychiatric Treatment*. 2008;14(2):152-158. doi:doi:10.1192/apt.bp.106.003400
4. Yang Y, Fang F, Arnberg FK, et al. Sex differences in clinically diagnosed psychiatric disorders over the lifespan: a nationwide register-based study in Sweden. *Lancet Reg Health Eur*. Dec 2024;47:101105. doi:10.1016/j.lanepe.2024.101105
5. Pasmán JA, Meijssen JJ, Haram M, et al. Epidemiological overview of major depressive disorder in Scandinavia using nationwide registers. *Lancet Reg Health Eur*. Jun 2023;29:100621. doi:10.1016/j.lanepe.2023.100621
6. Garcia-Argibay M, Brikell I, Thapar A, et al. Attention-Deficit/Hyperactivity Disorder and Major Depressive Disorder: Evidence From Multiple Genetically Informed Designs. *Biol Psychiatry*. Mar 1 2024;95(5):444-452. doi:10.1016/j.biopsych.2023.07.017
7. Bybjerg-Grauholm J, Bøcker Pedersen C, Bækvad-Hansen M, et al. The iPSYCH2015 Case-Cohort sample: updated directions for unravelling genetic and environmental architectures of severe mental disorders. *medRxiv*. 2020:2020.11.30.20237768. doi:10.1101/2020.11.30.20237768
8. Chow YY, Verdonschot M, McEvoy CT, Peeters G. Associations between depression and cognition, mild cognitive impairment and dementia in persons with diabetes mellitus: A systematic review and meta-analysis. *Diabetes Res Clin Pract*. Mar 2022;185:109227. doi:10.1016/j.diabres.2022.109227
9. Huang YY, Gan YH, Yang L, Cheng W, Yu JT. Depression in Alzheimer's Disease: Epidemiology, Mechanisms, and Treatment. *Biol Psychiatry*. Jun 1 2024;95(11):992-1005. doi:10.1016/j.biopsych.2023.10.008
10. Wightman DP, Jansen IE, Savage JE, et al. A genome-wide association study with 1,126,563 individuals identifies new risk loci for Alzheimer's disease. *Nat Genet*. Sep 2021;53(9):1276-1282. doi:10.1038/s41588-021-00921-z
11. Nguyen TD, Harder A, Xiong Y, et al. Genetic heterogeneity and subtypes of major depression. *Mol Psychiatry*. Mar 2022;27(3):1667-1675. doi:10.1038/s41380-021-01413-6
12. Nguyen TD, Kowalec K, Pasmán J, et al. Genetic Contribution to the Heterogeneity of Major Depressive Disorder: Evidence From a Sibling-Based Design Using Swedish National Registers. *Am J Psychiatry*. Oct 1 2023;180(10):714-722. doi:10.1176/appi.ajp.20220906